# Translaminar recurrence from layer 5 suppresses superficial cortical layers

Koun Onodera [1,2,3] & Hiroyuki K. Kato [1,2,4 ✉]

Information flow in the sensory cortex has been described as a predominantly feedforward sequence with deep layers as the output structure. Although recurrent excitatory projections from layer 5 (L5) to superficial L2/3 have been identified by anatomical and physiological studies, their functional impact on sensory processing remains unclear. Here, we use layer-selective optogenetic manipulations in the primary auditory cortex to demonstrate that feedback inputs from L5 suppress the activity of superficial layers regardless of the arousal level, contrary to the prediction from their excitatory connectivity. This suppressive effect is predominantly mediated by translaminar circuitry through intratelencephalic neurons, with an additional contribution of subcortical projections by pyramidal tract neurons. Furthermore, L5 activation sharpened tone-evoked responses of superficial layers in both frequency and time domains, indicating its impact on cortical spectro-temporal integration. Together, our findings establish a translaminar inhibitory recurrence from deep layers that sharpens feature selectivity in superficial cortical layers.

[1] Department of Psychiatry, University of North Carolina at Chapel Hill, Chapel Hill, NC, USA. [2] Neuroscience Center, University of North Carolina at Chapel Hill, Chapel Hill, NC, USA. [3] JSPS Overseas Research Fellow, Japan Society for the Promotion of Science, Tokyo, Japan. [4] Carolina Institute for Developmental Disabilities, University of North Carolina at Chapel Hill, Chapel Hill, NC, USA. ✉email: hiroyuki_kato@med.unc.edu

Revealing the principles governing interactions between the six layers of the cortex is fundamental for understanding how these intricately woven circuits work together to process sensory information. In the canonical cortical circuit model, information flow in the cortex has been considered as a predominantly feedforward sequence along the L4 → L2/3 → L5/6 axis, from which information is sent out to subcortical structures[1]. However, the cortex is not simply a feedforward circuit but is full of recurrence[2,3]. In recurrent circuits, the activity of even a single neuron can be transmitted back to itself, resulting in non-linear changes in the entire network. Recent studies have started to shed light on the computational consequences of local recurrence within a single cortical layer and have demonstrated its roles in shaping cortical sensory tuning[4–12]. In contrast, we still know little regarding how global recurrence encompassing multiple layers or brain regions affects cortical sensory processing. Given the strategical positioning of deep cortical layers that receive convergent inputs from ascending, descending, and lateral inputs[13–17], their feedback projections onto superficial layers could potentially expand the computational capacity of cortical circuits.

L5 of the sensory cortex is considered an output hub that sends long-range projections to a variety of other cortical and subcortical structures. L5 pyramidal cells are classified into two large categories[17–23]: intratelencephalic (IT) neurons, whose axons stay within the cortex and striatum, and pyramidal tract (PT) neurons, which project to multiple subcortical structures, including the thalamus. Importantly, IT neurons, and a subset of PT neurons, send their axons to superficial layers within the cortical column in addition to their distal targets[19], providing a potential substrate for translaminar recurrence between L2/3 and L5.

Indeed, this excitatory feedback projection from L5 pyramidal cells to L2/3 neurons has been confirmed by extensive anatomical and electrophysiological studies in vitro[2,24–30]. However, the functional impact of this excitatory feedback projection remains controversial in vivo. One study reported that activation of L5 pyramidal cells depolarized L2/3 neurons and triggered transitions to the global up state under anesthesia[31], supporting recurrent excitatory circuitry that enhances and prolongs sustained activity within the cortical column. In contrast, recent work combining two-photon calcium imaging and targeted optogenetics in awake mice found sparse activation of L5 neurons to be ineffective in driving L2/3 neurons, whereas activation of L2/3 neurons robustly recruited L5 neurons[32]. In agreement with the latter finding, L5 pyramidal cells also send projections to L2/3 inhibitory neurons, which may counteract the excitatory effects of L5 neurons onto superficial layers[2,26,33]. Therefore, it remains unclear how the interplay between excitatory and inhibitory pathways shapes the net functional impact of L5 neurons on sensory processing in superficial layers.

Here, by combining bidirectional optogenetic manipulations of L5 neurons with single-unit recordings across the entire cortical column in the primary auditory cortex (A1), we find the unexpected result that L5 activity in awake mice suppresses both spontaneous and tone-evoked activity in superficial layers. Thus, the ascending projection of L5 neurons forms an inhibitory recurrence, which sharpens tone-evoked responses of superficial layers in both the frequency and time domains. Recordings from the auditory thalamus and selective manipulations of L5 IT and PT subpopulations both indicate that this suppression is mediated by the coordinated action of intracortical and subcortical projections, with a substantial contribution of the intracortical inhibitory circuit. These results add a missing piece to existing cortical circuit models and reveal a critical role of translaminar recurrent circuits in shaping cortical sensory representations.

## Results

**Optogenetic activation of L5 suppresses superficial cortical layers.** To determine how L5 activity impacts sensory processing in other cortical layers, we conducted linear probe recordings across the entire cortical column while optogenetically manipulating L5 neurons in A1 of awake mice (Fig. 1a–c). We expressed the red light-sensitive cation channel ChrimsonR[34] in L5 pyramidal cells by injecting a Cre-dependent adeno-associated virus (AAV) vector in Rbp4-Cre mice. In this strain, transgene expression is restricted to excitatory neurons in both superficial and deep sublayers of L5[31,35,36] with dense neuropil in superficial layers (Fig. 1b). Three weeks after targeted AAV injections in A1 mapped with intrinsic signal imaging (Fig. 1d), recordings were conducted near the injection site by inserting a 64-channel linear probe perpendicular to the cortical surface. We recorded unit activity simultaneously from all six layers in awake head-fixed mice and identified L4 boundaries as the early sink in current source density analysis (Fig. 1e). Illumination of the A1 cortical surface with red LED (625 nm) triggered a rapid and reversible increase in the spontaneous firing of L5 regular-spiking (RS) units, demonstrating the effectiveness of our ChrimsonR-mediated photoactivation (Fig. 1f, g).

The prediction from the well-accepted L5 → L2/3 excitatory feedback connection is that activation of L5 will recruit L2/3 pyramidal cells into enhanced recurrent activity. To test this model, we first focused on RS units, which primarily represent excitatory neurons, and quantified the effect of L5 photoactivation by computing a modulation index (MI) for multi-unit firing at depth bins along the cortical column (Fig. 1g). MI was calculated as $(L - C)/(L + C)$, where $L$ represents the activity in LED trials and $C$ represents the activity during No LED control trials. Thus, MI ranges from -1 to 1, where a value of -1 represents a complete loss of activity, 1 represents the emergence of activity from nothing, and 0 represents no change. In contrast to our prediction, optogenetic L5 activation strongly suppressed spontaneous firing of RS units throughout the cortical column. Calculation of MI for isolated RS single-units demonstrated predominant suppression in both L2/3 and L4, while the effect on L6 was more variable (91%, 89%, and 63% of units showed suppression in L2/3, L4, and L6; L2/3: $p = 2.4 \times 10^{-11}$; L4: $p = 5.4 \times 10^{-10}$; L6: $p = 6.3 \times 10^{-5}$, Wilcoxon signed rank test; Fig. 1h, i, and Supplementary Fig. 1a). We observed only a small fraction of activated units, which mostly resided in L6 (6%, 5%, 27% of units in L2/3, L4, and L6; Supplementary Fig. 1a). Notably, suppression was stronger in L2/3 than L4 ($p = 0.016$, Wilcoxon rank sum test; Fig. 1i), suggesting that L2/3 does not simply inherit its suppression from L4. Thus, these data show that stimulation of L5 excitatory neurons suppresses activity in the superficial layers of A1.

A potential concern of photoactivation results is that the artificial L5 activity patterns may modulate cortical circuits differently from endogenous L5 activity. To address this issue and determine the impact of L5 activity on other layers in physiological conditions, we next performed loss-of-function experiments. We virally expressed inhibitory opsin eNpHR3.0 in L5 pyramidal cells using Rbp4-Cre mice. Illumination of the cortical surface with amber LED (595 nm) rapidly suppressed L5 RS unit activity in A1 (Fig. 1j, k). Consistent with our L5 photoactivation results, L5 inactivation enhanced the spontaneous firing of RS units throughout the cortical layers (Fig. 1k). MI for RS single-units demonstrated predominant activation of L2/3 and L4, with more variable effects on L6 (79%, 75%, and 52% of units showed enhancement in L2/3, L4, and L6; L2/3: $p = 6.1 \times 10^{-8}$; L4: $p = 9.3 \times 10^{-7}$; L6: $p = 0.13$, Wilcoxon signed rank test; Fig. 1l, m, and Supplementary Fig. 1b). Again, the

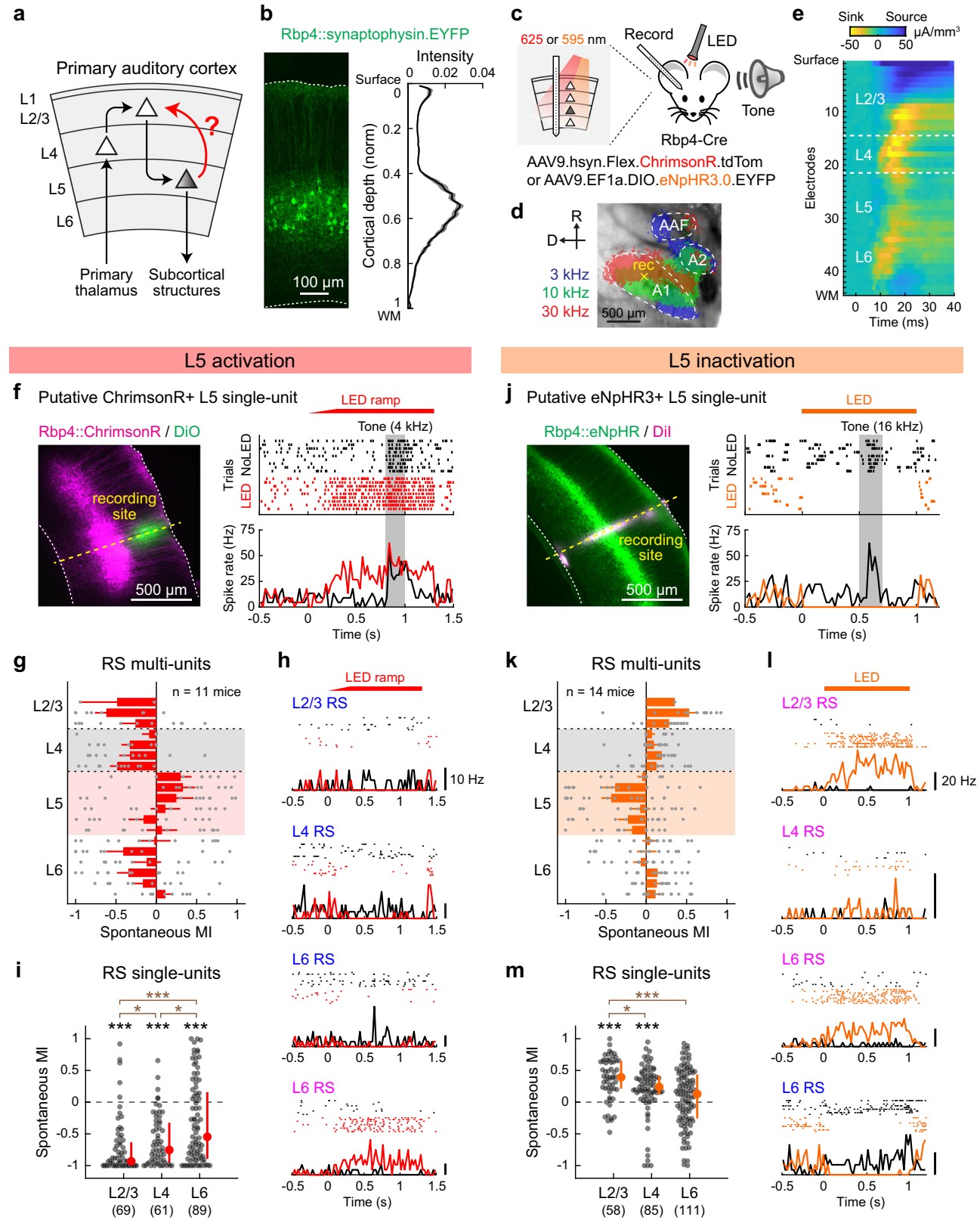

enhancement of activity was more pronounced in L2/3 than L4 ($p = 0.018$, Wilcoxon rank sum test; Fig. 1m), suggesting a site of suppression within L2/3. Collectively, our bidirectional manipulations demonstrate that both photoevoked and endogenous L5 activities have suppressive effects on superficial cortical layers in A1 of awake mice.

Our finding of L5-mediated negative regulation onto superficial layers contrasts with a previous study that reported L5 activation in anesthetized mice triggered a transition to cortical up state and increased L2/3 activity[31]. We wondered if this discrepancy could be due to different cortical circuit operations across brain states in recorded animals. To test the robustness of

**Fig. 1 Optogenetic activation of L5 suppresses superficial layers. a** Schematic illustrating the sensory information flow across cortical layers. **b** Left, coronal section of A1 from an Rbp4-Cre mouse injected with Cre-dependent AAV for synaptophysin-EYFP expression. Dotted lines indicate the upper and lower boundaries of the cortex. Right, distribution of EYFP signal intensity across the normalized cortical depth ($n = 3$ mice, 5 injection sites). Solid line, mean; shading, SEM. WM, white matter. **c** Schematics for optogenetics during linear probe recording across the cortical column. **d** Intrinsic signal imaging of responses to pure tones superimposed on cortical surface imaged through the skull. Yellow cross represents an A1 recording site. **e** Laminar profile of click sound-triggered current source density amplitudes in a representative A1 recording. Dotted lines indicate the identified borders of L4. **f–i** Data for L5 photoactivation with ChrimsonR. **f** Left, coronal section of A1 from mice expressing ChrimsonR-tdTomato in L5 neurons. The recording site (yellow dotted line) was identified by DiO signal. Right, raster and peristimulus time histogram (PSTH) of tone-evoked spikes in a photoactivated single-unit with (red) and without (black) LED. LED and No LED trials were interleaved but are separated for clarity. Red bar, 625 nm illumination; gray shading, tone stimulation at the unit's best frequency. **g** Laminar profile of modulation index (MI) for the spontaneous firing of regular-spiking (RS) multi-units during L5 activation, overlaid with individual data points ($n = 11$ mice). Depth is normalized in each mouse according to the surface, L4 borders, and white matter positions. Red shade indicates L5. Results are mean ± SEM. **h** Rasters and PSTHs of representative four single-units across layers. Blue letters indicate suppressed units, and pink letters indicate an enhanced unit. **i** Scatter plots showing MI for the spontaneous firing of RS single-units in each layer across mice (11 mice; $n = 69, 61, 89$ units for L2/3, L4, L6). Red dots and bars represent median and 25th and 75th percentiles. *$p < 0.05$, ***$p < 0.001$. Individual groups: two-sided Wilcoxon signed rank test with Bonferroni correction (black asterisks). Comparisons across groups: two-sided Wilcoxon rank sum test with Bonferroni correction (brown asterisks). **j–m** Same as in **f–i** but for L5 inactivation with eNpHR3 (14 mice; $n = 58, 85, 111$ units for L2/3, L4, L6). See Supplementary Data 1 for additional statistics. Source data are provided as a Source Data file.

our results across brain states, we examined if the arousal level of awake mice influenced the suppressive effects of L5 onto other layers. Throughout A1 unit recordings and L5 optogenetic manipulations, we monitored pupil diameter, a reliable readout for the animal's arousal level[37,38] (Fig. 2a). We used moderate LED intensities to prevent activity from reaching a floor or ceiling. As reported previously[37], L2/3 RS multi-unit spontaneous firing exhibited a U-shaped relationship with pupil size in the control condition (Fig. 2b). Bidirectional optogenetic manipulation of L5 activity scaled L2/3 firing rate while maintaining its U-shaped relationship with the pupil diameter, resulting in comparable MI values across arousal levels (L5 activation: $p = 0.96$; L5 inactivation: $p = 0.43$; one-way ANOVA; Fig. 2b±e). These results demonstrate that the suppressive effect of L5 onto superficial layers is robust regardless of the arousal level.

We next directly compared the effects of L5 optogenetic manipulations before and after inducing urethane anesthesia within the same mice (Fig. 2f). Similar to our results in the awake state, calculation of MI showed that L5 activation under anesthesia strongly suppressed spontaneous firing in superficial layers (96%, 80%, and 58% of single-units showed suppression in L2/3, L4, and L6; L2/3: $p = 8.4 \times 10^{-5}$; L4: $p = 3.7 \times 10^{-4}$; L6: $p = 0.55$, Wilcoxon signed rank test; Fig. 2g, h, and Supplementary Fig. 1c). The strength of suppression in L2/3–L4 and the increase of L5 firing were not different between the two states (L2/3–L4: $p = 0.67$; L5: $p = 0.45$, paired $t$-test; Fig. 2i). Similarly, L5 inactivation facilitated the activity of superficial layers under anesthesia (73%, 71%, and 68% of single-units showed enhancement in L2/3, L4, and L6; L2/3: $p = 0.018$; L4: $p = 0.0067$; L6: $p = 7.0 \times 10^{-4}$; Fig. 2j, k, and Supplementary Fig. 1d), and the strength of modulation was equivalent between the awake and anesthetized states (L2/3–L4: $p = 0.91$; L5: $p = 0.55$; Fig. 2l). We also confirmed that the use of constant-intensity LED without a ramp, similar to the one used in the previous study, or the use of weaker LED intensities did not affect our conclusions (Supplementary Fig. 2). Taken together, L5 activity suppresses superficial cortical layers regardless of brain state, indicating continuous sparsening of L2/3 neural activity by negative feedback from L5.

**L5 sharpens sensory tuning of superficial layers in both frequency and time domains.** How does the suppressive effect of L5 shape sound representations in superficial cortical layers? To address this question, we determined whether L5 activation alters frequency tuning of A1 neurons across the depth of the cortical column. We generated isointensity functions of RS single-units by

presenting pure tones across a range of frequencies (70 dB SPL, 4–64 kHz, 0.2 s), alternating control trials with L5 photostimulation trials (Fig. 3a). On top of the suppression of spontaneous firing, L5 activation attenuated tone-evoked activity in the superficial layers (Fig. 3a, b). Calculation of MI for tone-evoked activity showed that most RS single-units in L2/3 and L4 decreased their response amplitudes, while the effect on L6 was variable (78%, 67%, and 46% of unit-frequency pairs showed suppression in L2/3, L4, and L6; L2/3: $p = 3.4 \times 10^{-9}$; L4: $p = 4.9 \times 10^{-6}$; L6: $p = 0.12$, Wilcoxon signed rank test; Fig. 3c and Supplementary Fig. 3a). Similar to the impact on spontaneous activity, the suppression of tone responses was more robust in L2/3 than L4 ($p = 0.035$, Wilcoxon rank sum test). Comparison of the best frequency (BF) between control and LED trials showed that L5 photostimulation did not affect preferred frequency in other layers (Supplementary Fig. 3b). However, the overall suppression of tone-evoked activity narrowed the frequency tuning bandwidth in L2/3 and L4 (L2/3: $p = 1.7 \times 10^{-4}$; L4: $p = 0.018$; L6: $p = 0.16$, Wilcoxon signed rank test; Fig. 3d), indicating the sharpening of frequency tuning in superficial layers.

To understand the transformation underlying L5-mediated sharpening of frequency tuning in L2/3–L4 neurons, we distinguished between subtractive and divisive mechanisms by plotting tone-evoked responses under control versus L5 photostimulation conditions (Fig. 3e)[39–41]. By applying threshold-linear fit to the data points, we determined whether suppression in individual single-units was subtractive (y-intercept far from zero), divisive (slope far from one), or a combination of both (Methods; Supplementary Fig. 4). We found that suppression in L2/3–L4 was a mixture of subtractive and divisive control, with the latter being dominant (Fig. 3f, g): approximately half (42%) of units showed only divisive suppression, 9% showed only subtractive suppression, and 15% showed both mechanisms. This result is distinct from the purely divisive scaling found in L6-dependent suppression in the primary visual cortex (V1)[42] but consistent with the mixed mechanisms of gain regulation by A1 cortical inhibitory neuron activation[39].

In L5 inactivation experiments, we observed the overall inverse of the L5 activation results, except for less pronounced effects on L4 (see Discussion). Calculation of MI showed that most RS single-units in L2/3 increased their tone-evoked responses, while L4 and L6 results were variable (57%, 42%, and 24% of unit-frequency pairs showed enhancement in L2/3, L4, and L6; L2/3: $p = 9.1 \times 10^{-11}$; L4: $p = 0.20$; L6: $p = 0.51$, Wilcoxon signed rank test; L2/3 vs. L4: $p = 0.0040$, Wilcoxon rank sum test; Fig. 3h–j and Supplementary Fig. 3c). The overall enhancement of tone-

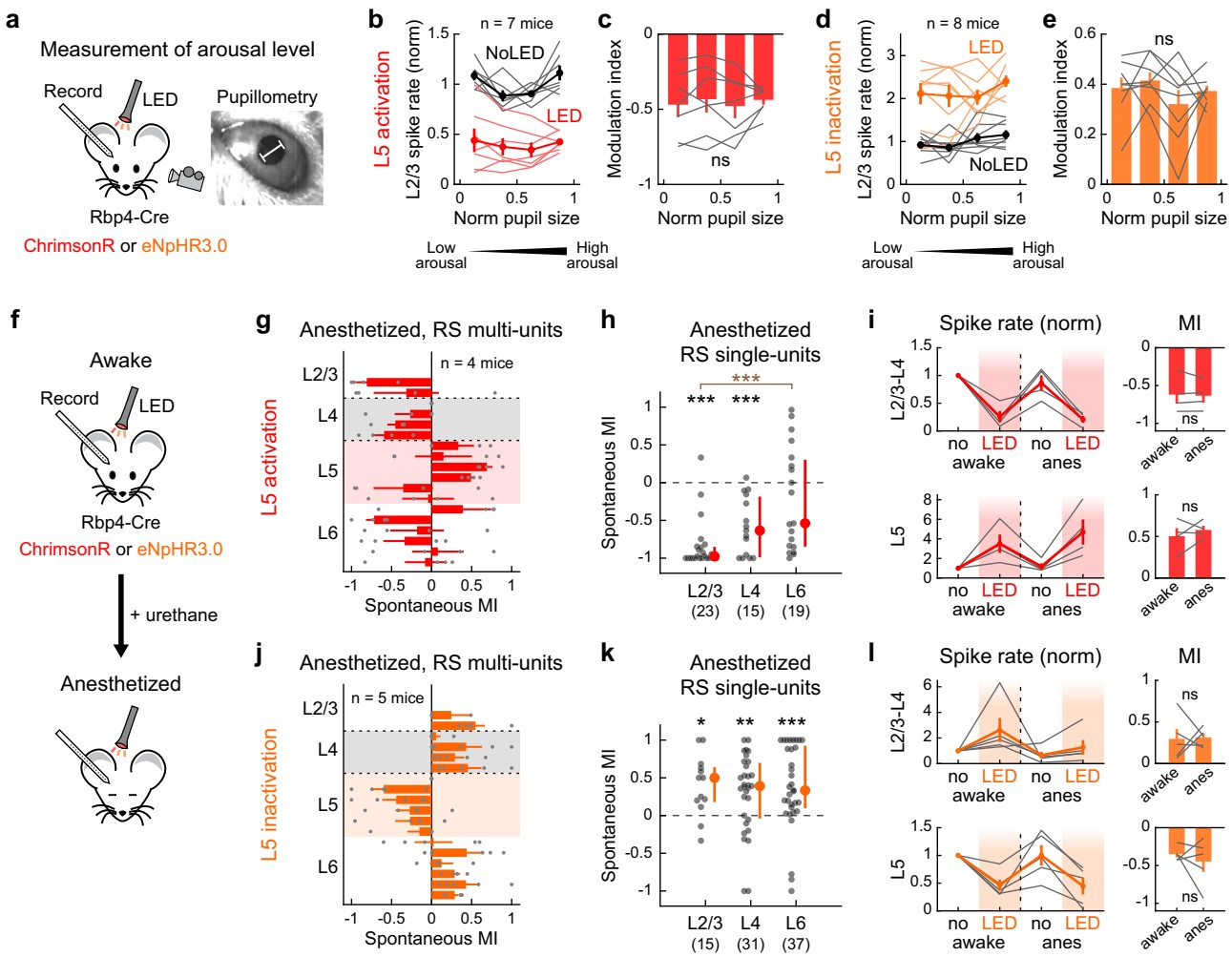

**Fig. 2 Optogenetic activation of L5 suppresses superficial layers regardless of brain states. a** Left, schematics for the experimental setup with pupil monitoring. Right, representative pupil camera image. White bar indicates pupil diameter. **b** Dependence of L2/3 multi-unit spontaneous firing rate on pupil diameter with (red) and without (black) L5 activation (*n* = 7 mice). Thin lines show individual mice, and thick lines indicate mean ± SEM. Firing rate is normalized to that of No LED trials. Data are grouped into four bins of arousal level based on normalized pupil diameter in each mouse. **c** Dependence of MI of L2/3 multi-unit activity on pupil diameter during L5 activation. Data are mean ± SEM, overlaid with individual mice (n = 7 mice). p = 0.96. One-way ANOVA. **d, e** Same as in **b**, **c** but for L5 inactivation (*n* = 8 mice). *p* = 0.43. **f** Schematics for A1 recordings in awake and anesthetized states. **g–i** Data for L5 activation with ChrimsonR under urethane anesthesia. **g** Laminar profile of MI for the spontaneous firing of RS multi-units during L5 activation, overlaid with individual data points (n = 4 mice). Red shade, L5. Results are mean ± SEM. **h** Scatter plots showing MI for the spontaneous firing of RS single-units in each layer across mice (4 mice; *n* = 23, 15, 19 for L2/3, L4, L6). Red dots and bars represent median and 25th and 75th percentiles. ***p < 0.001. Individual groups: two-sided Wilcoxon signed rank test with Bonferroni correction (black asterisks). Comparisons across groups: two-sided Wilcoxon rank sum test with Bonferroni correction (brown asterisks). **i** Left, spontaneous multi-unit firing rates normalized to no LED condition in the awake state separately for L2/3–L4 (top) and L5 (bottom). LED and No LED trials were interleaved in each brain state. Red line plot indicates mean ± SEM. Gray lines show individual mice. Right, summary plots comparing MI for spontaneous activity between awake and anesthetized states (n = 4 mice; L2/3–L4: p = 0.67; L5: p = 0.45; two-sided paired *t*-test). **j–l** Same as in **g–i** but for L5 inactivation with eNpHR3 under urethane anesthesia (5 mice, *n* = 15, 31, 37 for L2/3, L4, L6). *p < 0.05, **p < 0.01. See Supplementary Data 1 for additional statistics. Source data are provided as a Source Data file.

evoked activity broadened the tuning of L2/3 units without affecting their preferred frequency (L2/3: *p* = 0.0021; L4: *p* = 1.0; L6: *p* = 1.0, Wilcoxon signed rank test; Fig. 3k and Supplementary Fig. 3d). Application of threshold-linear fitting to each tone-responsive unit showed that the impact of L5 inactivation on superficial layers was a mixture of additive and multiplicative transformations, with the latter being slightly more dominant (Fig. 3l–n). Together, our results from bidirectional manipulations of L5 revealed that both photoevoked and endogenous activity of L5 sharpens frequency tuning in superficial cortical layers. The mixed divisive and subtractive transformations may indicate cortical inhibitory mechanisms[39] in L5-dependent suppressive gain control of superficial layers.

Recurrent excitatory circuits are considered to provide neural substrates for maintaining sustained activity[43]. Our unexpected finding of inhibitory recurrence from L5 onto L2/3 thus raises a question—does L5 activity prolong[31] or truncate[44] sensory activity in superficial layers? To address this question, we examined the impacts of L5 activation and inactivation on three kinetics parameters: response latency, decay time, and full-width of half-maximum (FWHM) of tone-evoked responses in L2/3–L4 multi-unit activity (Fig. 4a,e). While we found no change in response latency with either manipulation (activation: *p* = 0.13; inactivation: *p* = 0.10, paired *t*-test, Fig. 4b, f), L5 activation and inactivation shortened and prolonged decay time, respectively (activation: *p* = 0.074; inactivation: *p* = 0.0027, Fig. 4c, g).

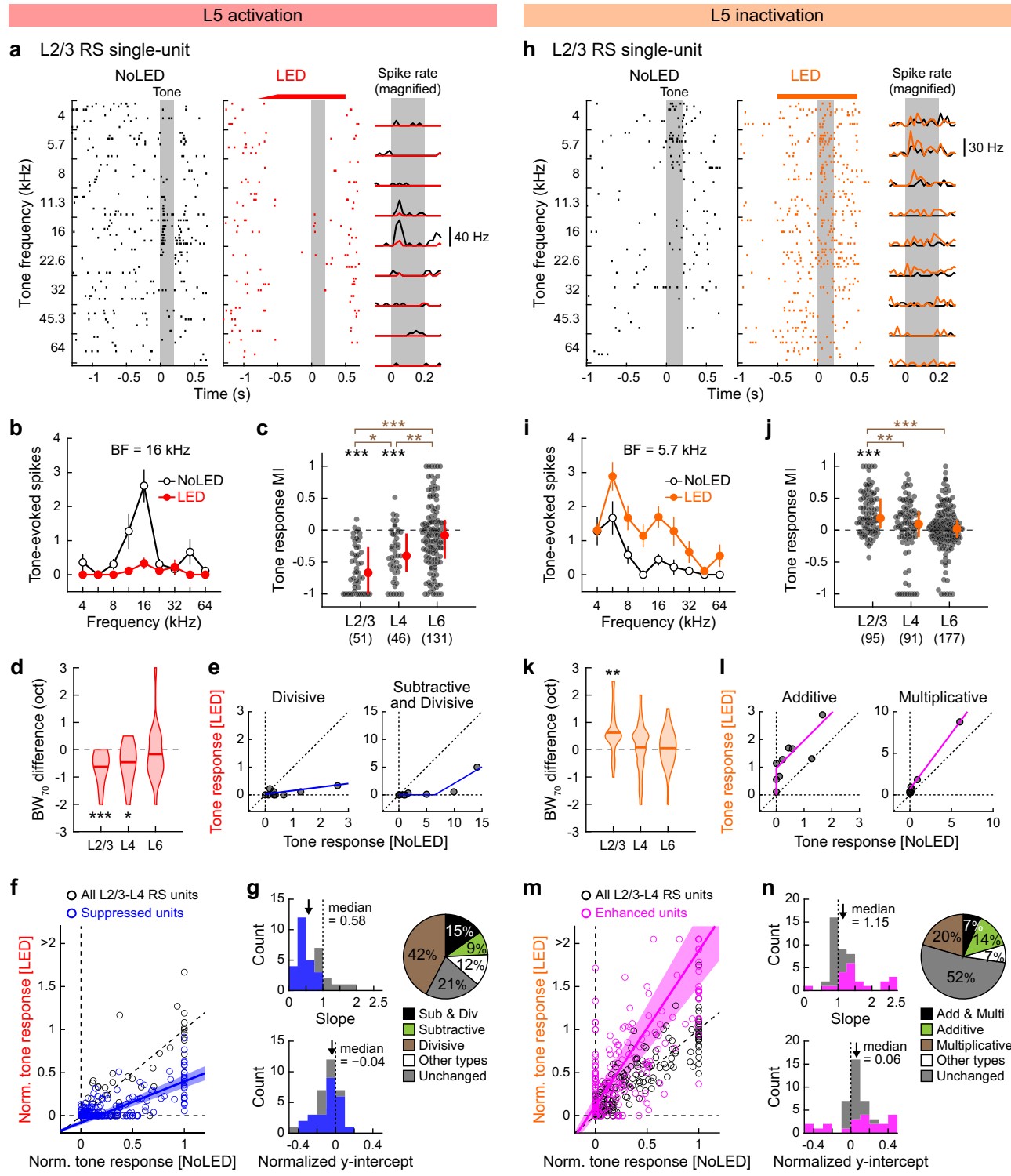

Consequently, we observed FWHM truncation in L5 activation ($p = 0.016$; Fig. 4d) and elongation in L5 inactivation ($p = 0.020$; Fig. 4h). Therefore, L5 activity sharpens the timing of tone-evoked responses in superficial layers, further supporting inhibitory rather than excitatory recurrence of L5 → L2/3 circuitry. Together, these results demonstrate that L5 sharpens sound responses of superficial layers in both the spectral and temporal dimensions, suggesting its impact on cortical spectro-temporal integration.

**Intracortical and subcortical pathways for L5 inhibitory recurrence**. Two independent pathways could mediate L5-

dependent suppression of superficial cortical layers. On one hand, intracortical projections of L5 neurons could recruit local inhibitory neurons to suppress superficial layers. On the other hand, L5 activity could indirectly suppress the primary auditory thalamus (ventral part of the medial geniculate nucleus: MGv), which may in turn reduce feedforward excitation onto A1 L4. In our optogenetic manipulation, the suppressive effect of L5 was consistently larger in L2/3 than its upstream L4 (Fig. 1i, m; Fig. 3c, j), indicating the role of intracortical mechanisms. To examine if there is an additional contribution of subcortical pathways, we evaluated the relative impacts of both mechanisms

**Fig. 3 Optogenetic activation of L5 sharpens the spectral tuning in superficial layers. a–h** Data for A1 tone-evoked responses during L5 activation with ChrimsonR. **a** Tone evoked responses of a representative L2/3 RS single-unit with (red) and without (black) L5 activation. Left and middle, raster plots of tone-evoked spikes for nine tone frequencies. LED and No LED trials were interleaved but are separated for clarity. Red bar, 625 nm illumination. Gray shading, tone. Right, PSTH. **b** Isointensity function for the unit shown in **a**. Data are mean ± SEM of nine trials. **c** Scatter plots showing MI for the tone-evoked firing of RS single-units in each layer across mice (11 mice; $n = 28, 24, 43$ tone-responsive units and 51, 46, 131 responsive unit-frequency pairs for L2/3, L4, L6). Red dots and bars represent median and 25th and 75th percentiles, respectively. $*p < 0.05$, $**p < 0.01$, $***p < 0.001$. Individual groups: two-sided Wilcoxon signed rank test with Bonferroni correction (black asterisks). Comparisons across groups: two-sided Wilcoxon rank sum test with Bonferroni correction (brown asterisks). **d** Violin plots showing LED-induced changes in the population frequency tuning bandwidth at 70 dB SPL (BW$_{70}$) in each layer. Thick lines indicate mean. Two-sided Wilcoxon signed rank test with Bonferroni correction. **e** Tone-evoked spike counts of representative L2/3 RS single-units with divisive (left; unit shown in **a**) or both divisive and subtractive (right) suppression with LED. Blue lines show fit with threshold-linear functions. Dotted lines indicate $x = 0$, $y = 0$, and unity. **f** Summary data showing well-fit RS single-units in L2/3–L4 ($n = 33$ single-units, 297 unit-tone pairs). Spike counts are normalized to the BF response without LED in each unit. Blue line and shading show mean and SEM of all linear fit lines of suppressed units. Suppressed units (blue circles) were defined by the MI of the total spike counts for all responsive frequencies; therefore, some of the data points for suppressed units could appear above the unity line. **g** Distribution of best-fit slope (top left) and normalized y-intercept (bottom left) for all L2/3–L4 RS single-units that were fit well with threshold-linear functions. Blue bars indicate units with significant suppression of tone-evoked activity. Arrows show median. Right, a pie chart showing the fraction of units with divisive (brown), subtractive (green), both suppression (black), other modulations (white), and no change (gray) ($n = 33$ units excluding units with poor fitting). **h–n** Same as in **a–g** but for L5 inactivation with eNpHR3. **i** Data are mean ± SEM of nine trials. **j** 14 mice; $n = 28, 30, 53$ tone-responsive units and 95, 91, 177 unit-frequency pairs for L2/3, L4, L6. **l** Tone-evoked spike counts of representative L2/3 RS single-units with additive (left; unit shown in **h**) or multiplicative (right) enhancement with LED. Pink lines show fit with threshold-linear functions. **m** $n = 44$ single-units, 396 unit-tone pairs. Pink line and shading show mean and SEM of all linear fit lines of enhanced units. **n** $n = 44$ units that were fit well with threshold-linear functions. See Supplementary Data 1 for additional statistics. Source data are provided as a Source Data file.

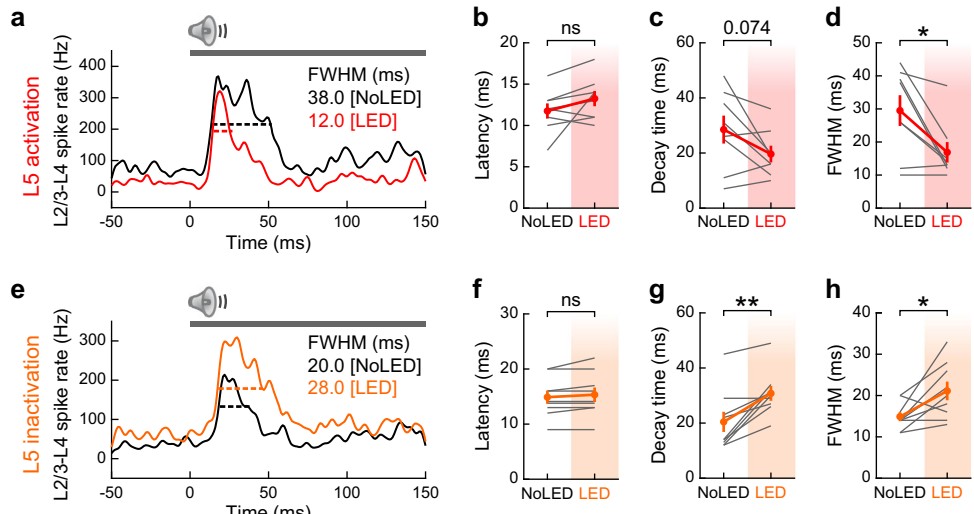

**Fig. 4 Photoactivation of L5 sharpens spike timing in superficial layers. a** PSTH of tone-evoked multi-unit activity combining L2/3–L4 spikes with (red) or without (black) L5 activation in a representative mouse. Black and red bars show full width at half maximum (FWHM). Gray bar, tone stimulation. **b–d** Summary data showing **b** onset latency (time to 20% maximum), **c** decay time (time from 100% to 30% maximum), and **d** FWHM, for all mice ($n = 8$ mice; latency, $p = 0.13$; decay, $p = 0.074$; FWHM, $*p = 0.016$; two-sided paired $t$-test). Data are mean ± SEM. Frequencies with significant responses in either LED or No LED conditions were included in multi-unit spikes. Only multi-unit data with significant responses in both LED and No LED conditions were included in kinetics analyses. These necessary selection criteria biased FWHM towards larger values in L5 activation experiments since mice with small responses in No LED trials tended to be excluded due to the loss of responsiveness in LED trials. **e–h** Same as in **a–d** but for L5 inactivation with eNpHR3 ($n = 9$ mice; latency, $p = 0.10$; decay, $**p = 0.0027$; FWHM, $*p = 0.020$; two-sided paired $t$-test). Data are mean ± SEM. See Supplementary Data 1 for additional statistics. Source data are provided as a Source Data file.

by conducting unit recordings in A1 and MGv in the same mice during L5 manipulation (Fig. 5a). By inserting a linear probe deeper after A1 recordings, we were able to reach MGv, where we observed time-locked click sound responses (Fig. 5b, c). We found that L5 optogenetic activation reduced spontaneous spikes in MGv, but to a lesser extent than in A1 L2/3–L4 (MGv: 40.2% reduction; L2/3-L4: 59.9%; $p = 0.033$, paired $t$-test; Fig. 5d, e). In the L5 inactivation experiments, the difference of modulation magnitudes between MGv and A1 superficial layers was more prominent; L5 inactivation only slightly increased spontaneous spikes in MGv, whereas A1 L2/3–L4 activity showed four-times

larger enhancement in the same mice (MGv: 19.2% enhancement; L2/3-L4: 79.5%; $p = 0.028$, paired $t$-test; Fig. 5f, g). These results indicate the involvement of both intracortical and subcortical pathways in L5-dependent suppression of superficial cortical layers, although the endogenous activity of L5 acts predominantly through intracortical mechanisms. In a separate set of mice, we also performed unit recordings in the inferior colliculus. Consistent with a previous study[45], L5 activation triggered a small enhancement of spontaneous firing in the external cortex, but not the central nucleus of the inferior colliculus (Supplementary Fig. 5). The lack of effect on the lemniscal inferior colliculus

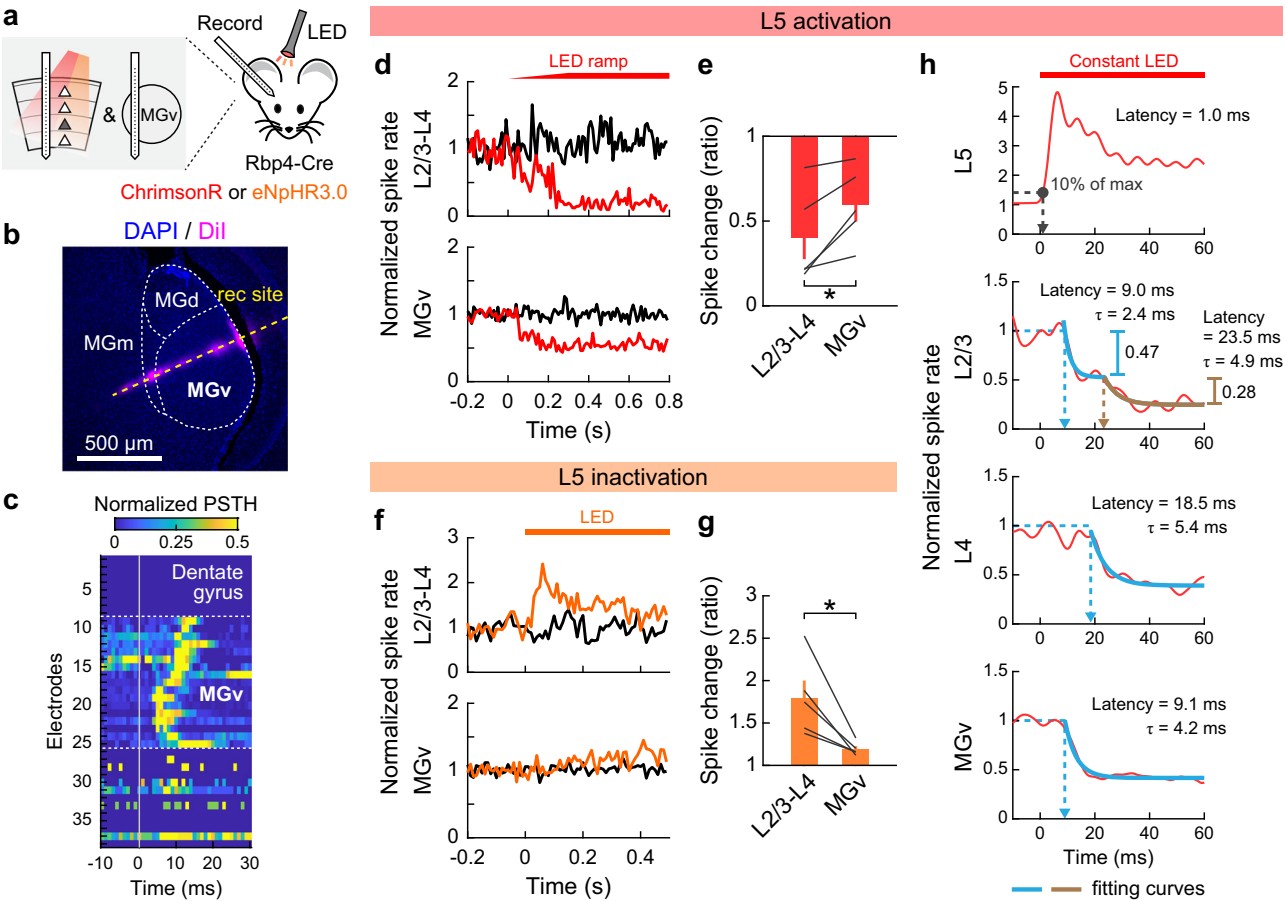

**Fig. 5 L5 suppresses L2/3 via both intracortical and subcortical mechanisms. a** Schematics for successive recordings of A1 and MGv in the same animal with optogenetic manipulation of L5. **b** Coronal section of auditory thalamus. Recording site (yellow dotted line) was identified by DiI signal. **c** Heat map showing normalized PSTH of click sound-evoked firing around MGv. PSTH was normalized to the maximum value in each channel to visualize the position dependence of spike latencies. Dotted lines indicate the identified borders of MGv. **d**, **e** Data for L5 activation with ChrimsonR. **d** Normalized PSTHs of multi-unit activity combining spikes in L2/3–L4 (top) and MGv (bottom) with (red) and without (black) L5 activation in a representative mouse. Red bar, 625 nm LED. **e** Summary data comparing LED-dependent suppression of spontaneous firing between L2/3–L4 and MGv ($n = 5$ mice; *$p = 0.033$, two-sided paired $t$-test). Data are mean ± SEM. **f**, **g** Same as **d**, **e** but for L5 inactivation with eNpHR3 ($n = 5$ mice). *$p = 0.028$. **h** Time course of spontaneous activity changes triggered by L5 activation using constant LED illumination without a ramp (red bar). Multi-unit activity in each layer was combined across mice after removing photoactivated units in L2/3, L4 ($n = 6$ mice), and MGv ($n = 4$ mice). Gray dot, delay for 10% maximum activation in L5. Blue lines, first exponential fit curves. Brown line, second exponential fit curve. Fitting of L4 and MGv data with two exponential curves returned negligible second components. See Supplementary Data 1 for additional statistics. Source data are provided as a Source Data file.

suggests that the L5-dependent small suppression of MGv is not inherited from upstream and arises at the level of the thalamus.

Intracortical and subcortical mechanisms of L5-dependent inhibitory recurrence should suppress the superficial layers with distinct kinetics. To test this prediction, we combined multi-unit spikes during L5 optogenetic activation across mice and compared the suppression kinetics between A1 L2/3, L4, and MGv. Note that we excluded L2/3–L4 units with increased spikes to clarify the suppression onset; however, the incomplete removal of activated units likely caused an overestimation of suppression latency in these layers. Despite this overestimation, we found that the suppression onset for L2/3 was earlier than L4 and comparable to MGv (9.0, 18.5, and 9.1 ms latency in L2/3, L4, and MGv; Fig. 5h), indicating that MGv suppression was not fast enough to account for the short-latency suppression in L2/3. Indeed, the L2/3 spike rate was fit the best by the sum of two exponential curves, which likely corresponds to fast intracortical and slow subcortical mechanisms, while L4 and MGv data were fit better by a single exponential. Quantification of suppression magnitudes explained by the first and second exponentials

revealed that the fast component accounted for 63% of total suppression in L2/3 (fast component: 0.47; slow component: 0.28). Together, these data provide further evidence that L5 activity suppresses L2/3 neurons predominantly through the intracortical pathway.

**IT neurons modulate superficial layers more robustly than PT neurons**. Our results revealed the contribution of both intracortical and subcortical pathways in L5-mediated inhibitory recurrence onto superficial cortical layers. Based on this observation, we next asked if there are distinct roles for the two large categories of L5 pyramidal cells: IT neurons, which predominantly contribute to intracortical projections, and PT neurons, which are the sole source of subcortical projections (Fig. 6a). To address this question, we selectively manipulated L5 IT and PT neurons using a combination of viral approaches. Since IT neurons connect to PT neurons while PT neurons rarely synapse onto IT neurons, the difference in the effects between the two manipulations is likely attributed to intracortical projections of IT neurons.

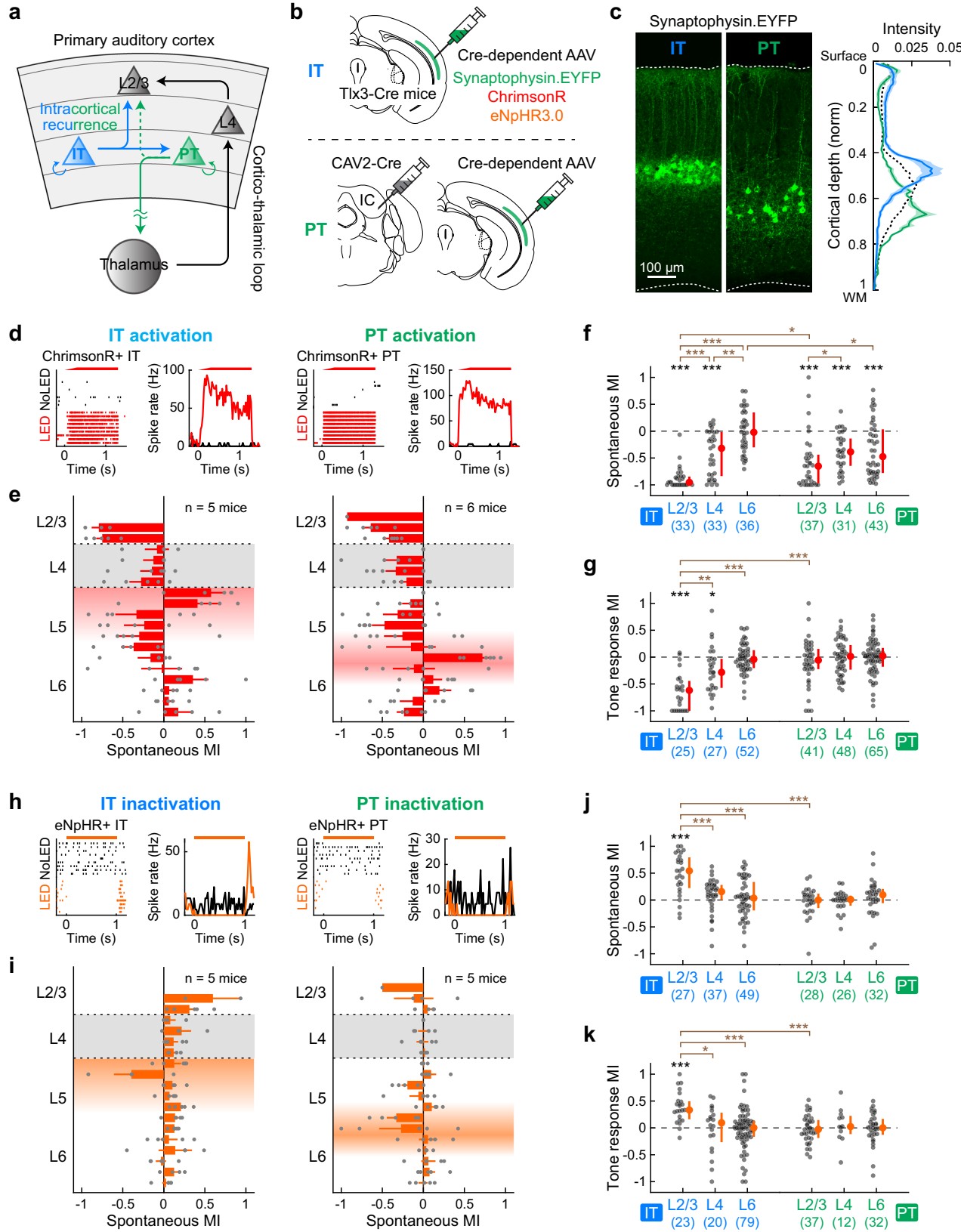

We selectively expressed optogenetic tools in IT neurons using Tlx3-Cre mice[17,35] (Fig. 6b). PT neurons were retrogradely labeled by injection of CAV2-Cre into the inferior colliculus[16,46]. By expressing synaptophysin-EYFP using these strategies, we confirmed distinct distribution of their somata across the cortical depth (IT neurons in L5a and PT neurons in L5b; Fig. 6c) and

their differential projections onto subcortical structures (Supplementary Fig. 6).

To determine the difference between the effects of IT and PT activation, we first expressed ChrimsonR selectively in each subtype and conducted photostimulation during A1 unit recordings. Illumination of the cortical surface with red LED

**Fig. 6 L5 IT neurons modulate superficial layers more robustly than PT neurons. a** Schematics showing potential feedback circuits via IT and PT neurons in L5. **b** Viral strategies for gene expression selectively in IT (top) or PT(bottom) neurons. **c** A1 coronal sections from mice expressing synaptophysin-EYFP in IT (left) or PT (middle) neurons. Right, distribution of EYFP signal intensity across the normalized cortical depth (IT, $n = 3$ mice, 5 sections; PT, $n = 4$ mice, 5 sections). Solid lines and shadings show mean and SEM. Black dotted line shows the mean of Rbp4-Cre data from Fig. 1b. **d–g** Data for activation of IT or PT neurons with ChrimsonR. **d** Left, raster plot and PSTH of spontaneous firing with (black) or without (red) LED in a representative photoactivated IT single-unit. Right, similar plots for a representative PT single-unit. **e** Laminar profile of MI for the spontaneous firing of RS multi-units during activation of IT (left) or PT (right) neurons. Individual data points are overlaid (IT, $n = 5$ mice; PT, $n = 6$ mice). Depth is normalized in each mouse according to the surface, L4 borders, and white matter positions. Red shades indicate L5a (left) and L5b (right), respectively. Results are mean ± SEM. **f** Scatter plots showing MI for the spontaneous firing of RS single-units in each layer across mice (IT: $n = 33, 33, 36$ units; PT: $n = 37, 31, 43$ units for L2/3, L4, L6). Red dots and bars represent median and 25th and 75th percentiles, respectively. $*p < 0.05$, $**p < 0.01$, $***p < 0.001$. Individual groups: two-sided Wilcoxon signed rank test with Bonferroni correction (black asterisks). Comparisons across groups: two-sided Wilcoxon rank sum test with Bonferroni correction (brown asterisks). **g** Same as **f** for tone-evoked firing (IT: $n = 11, 12, 11$ tone-responsive units and $n = 25, 27, 52$ unit-frequency pairs for L2/3, L4, L6; PT: $n = 15, 17, 19$ tone-responsive units and $n = 41, 48, 65$ unit-frequency pairs for L2/3, L4, L6). **h–k** Same as in **d–g** but for inactivation of IT or PT neurons with eNpHR3. **i** IT: $n = 5$ mice; PT: $n = 5$ mice. **j** IT: $n = 27, 37, 49$ units for L2/3, L4, L6; PT: $n = 28, 26, 32$ units for L2/3, L4, L6. **k** IT: $n = 10, 13, 28$ tone-responsive units and $n = 23, 20, 79$ unit-frequency pairs for L2/3, L4, L6; PT: $n = 16, 6, 11$ tone-responsive units and $n = 37, 12, 32$ unit-frequency pairs for L2/3, L4, L6. See Supplementary Data 1 for additional statistics. Source data are provided as a Source Data file.

triggered an increase in spontaneous firing at expected cortical depths (Fig. 6d, e), and the magnitudes of L5 activation were comparable between IT and PT manipulations (Supplementary Fig. 7). Calculation of MI for RS single-units demonstrated two noticeable differences between IT and PT photostimulation. First, although both IT and PT activation suppressed L4 spontaneous activity to a comparable level (IT vs. PT: $p = 1.0$, Wilcoxon rank sum test), IT activation triggered significantly larger suppression in L2/3 (IT vs. PT: $p = 0.021$; Fig. 6f and Supplementary Fig. 8), further supporting the suppression of L2/3 by the L5 intracortical projections. PT activation additionally suppressed L6, which was absent in IT activation (IT vs. PT: $p = 0.014$). Second, MI for tone-evoked activity demonstrated that only IT, not PT, activation suppressed tone responses in superficial layers (L2/3: $p = 1.3 \times 10^{-4}$; L4: $p = 0.040$, Wilcoxon signed rank test; Fig. 6g). Indeed, the sharpening of frequency tuning in superficial layers was comparable between pan-L5 neuron activation and IT neuron activation alone (Fig. 3 and Supplementary Fig. 9). Therefore, intracortical recurrence from IT neurons plays a crucial role in regulating both spontaneous and tone-evoked activity of superficial layers. In contrast, subcortical projections from PT neurons suppress only spontaneous firing and do not affect tone-evoked activity.

We next expressed eNpHR3.0 selectively in IT or PT neurons and conducted subtype-specific loss-of-function experiments (Fig. 6h, i). Surprisingly, although the magnitudes of L5 inactivation were again comparable between IT and PT manipulations (Supplementary Fig. 7), quantification of MI for RS single-units revealed a striking difference. Inactivation of IT neurons enhanced both spontaneous and tone-evoked activity in L2/3 (spontaneous: $p = 2.9 \times 10^{-4}$; tone-evoked: $p = 4.1 \times 10^{-4}$, Wilcoxon signed rank test; Fig. 6j, k), similar to pan-L5 inactivation using Rbp4-Cre mice (Fig. 1m, 3j, and Supplementary Fig. 9). In contrast, PT inactivation did not cause any change in the firing of other layers. The lack of effects by PT photoinactivation agrees with our observation that pan-L5 inactivation minimally affected MGv activity (Fig. 5f, g), which both indicate that the endogenous level of L5 activity is insufficient in suppressing subcortical structures. Taken together, our results demonstrate that intracortical projections of IT neurons predominantly mediate the suppressive effects of L5 onto superficial layers. Subcortical projections of PT neurons likely provide an additional layer of suppression only in the face of substantially raised activity.

Finally, we investigated the inhibitory mechanisms by which intracortical L5 recurrence negatively regulates superficial layers. To test whether changes in inhibitory neuron activity account for

these suppressive effects, we combined data from pan-L5 (Fig. 1) and IT neuron (Fig. 6) manipulation experiments and analyzed fast-spiking (FS) units, which mostly represent parvalbumin-expressing inhibitory neurons. Interestingly, optogenetic activation of L5 neurons suppressed FS single-units in L2/3–L4 while those in L5–L6 were variable but overall unchanged (Fig. 7a, b). Therefore, the suppressive effects of L5 cannot be simply explained by the increased activity of inhibitory neurons. However, focusing on spike changes at the LED onset, we found many L2/3–L4 FS units that transiently increased their firing before switching to suppression (Fig. 7c, d). Combining transient and sustained activation, we found a three times higher fraction of FS single-units showing LED-driven excitation than RS single-units (RS: 11.2%; FS: 39.2%; $p = 0.0015$; Fisher exact test). The biphasic modulation of inhibitory neurons and co-suppression of both excitatory and inhibitory neurons are in agreement with the recently observed rebalancing of cortical circuits after optogenetic manipulation of inhibitory neurons in awake mice[6,7,10], which theoretical studies have attributed to the dynamics of the local recurrent network (inhibition-stabilized network model; Supplementary Fig. 10)[4,8,47]. Conversely, optogenetic inactivation of L5 neurons increased the firing of L2/3–L4 FS units without affecting those in L5–L6 (Fig. 7e, f). We observed biphasic modulations of L2/3–L4 FS units at the LED onset and found a three times higher fraction of FS single-units showing LED-driven suppression than RS units (RS: 9.7%; FS: 30.0%; $p = 0.016$; Fisher exact test; Fig. 7h), which again agrees with circuit rebalancing. Therefore, our data are most consistent with a model where L5 IT neurons send ascending projections that preferentially synapse onto FS inhibitory neurons in superficial layers, although we do not exclude the contribution of other inhibitory neuron subtypes.

## Discussion

Dense recurrent networks, including feedforward, feedback, and lateral projections, are a hallmark of mammalian cortical circuits and are considered to underlie higher integrative functions, such as working memory and consciousness[43,48–50]. Both anatomical and electrophysiological studies have found that recurrent circuits provide the majority of sensory-evoked excitation to cortical neurons[2,11,51,52]. Therefore, understanding cortical computations critically relies on our knowledge of the functions of recurrent circuits. While extensive previous work has described the mutual excitatory projections between L2/3 and L5 pyramidal cells in many cortical areas, the functional consequence of translaminar recurrent circuits is more elusive. In this study, we took advantage of genetic and viral tools to selectively manipulate L5 pyramidal

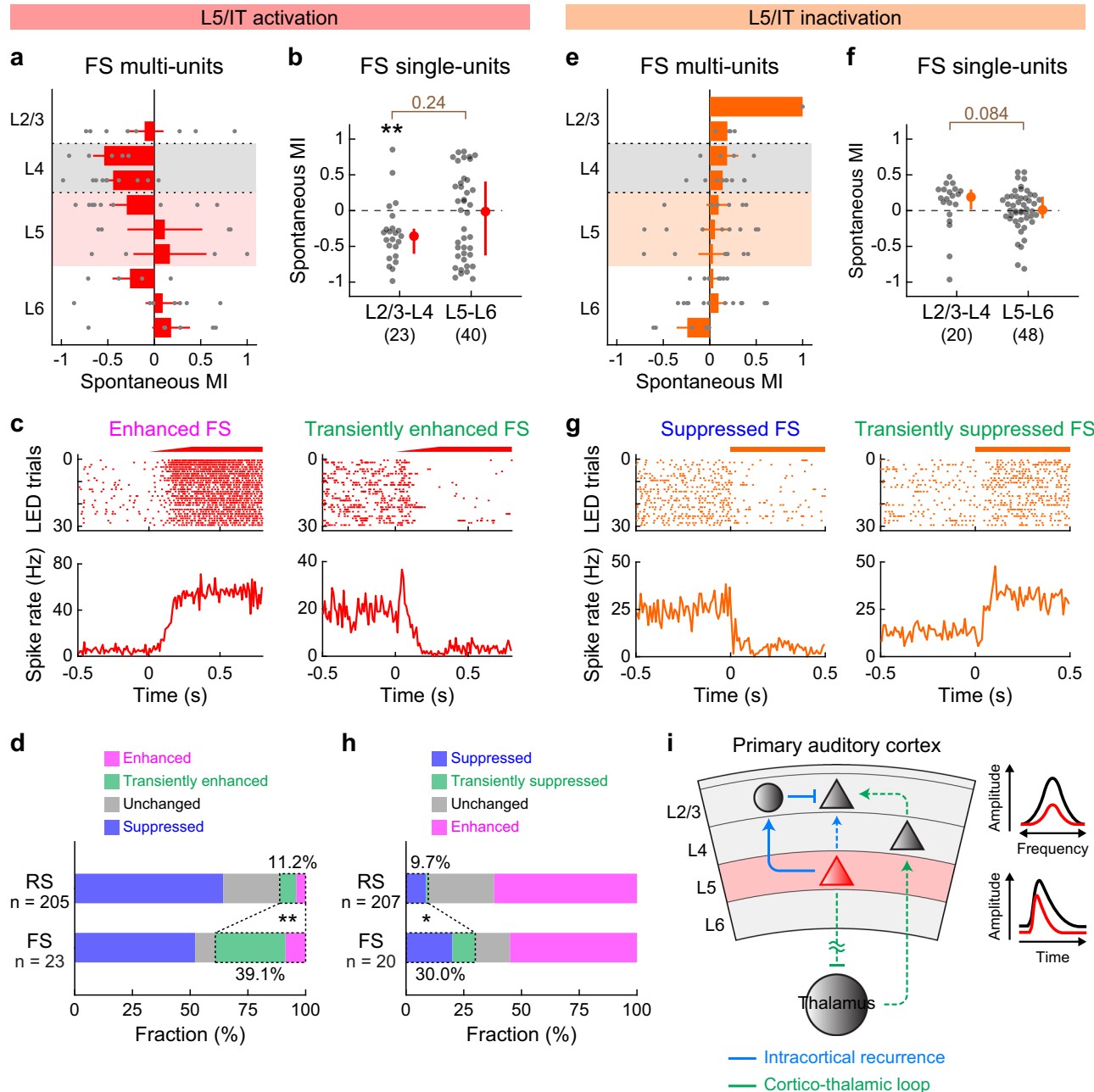

**Fig. 7 L5 recurrent inputs preferentially recruit fast-spiking units in superficial layers. a–d** Data for fast-spiking (FS) units during L5 activation with ChrimsonR, combining data from Rbp4-Cre and Tlx3-Cre mice. **a** Laminar profile of MI for the spontaneous firing of FS multi-units during L5 activation overlaid with individual data points (n = 16 mice). Depth is normalized in each mouse according to the surface, L4 borders, and white matter positions. Red shade indicates L5. Larger bins were used compared to RS units data due to the sparsity of FS units. Results are mean ± SEM. **b** Scatter plot showing MI for the spontaneous firing of FS single-units across mice (n = 23 and 40 for L2/3-L4 and L5-L6). Red dots and bars represent median and 25th and 75th percentiles. **p < 0.01. Individual groups: two-sided Wilcoxon signed rank test with Bonferroni correction (black asterisks). Comparisons across groups: two-sided Wilcoxon rank sum test (brown). **c** Raster and PSTH of representative L2/3 FS single-units showing sustained (left) or transient (right) enhancement of spontaneous firing during L5 activation. **d** Fraction of L2/3-L4 RS and FS single-units showing sustained enhancement (pink), transient enhancement (green), suppression (blue), and no change (gray) (16 mice; RS, n = 205 units; FS, n = 23 units). **p = 0.0015 (two-sided Fisher exact test). **e–h** Same as in **a–d** but for L5 inactivation with eNpHR3. **e** n = 19 mice. **f** n = 20 and 48 for L2/3-L4 and L5-L6. **g** Raster and PSTH of representative FS single-units showing sustained (left) or transient (right) suppression of spontaneous firing during L5 inactivation. **h** n = 19 mice; RS, n = 207 units; FS, n = 20 units. *p = 0.016 (two-sided Fisher exact test). **i** Schematic illustrating the translaminar inhibitory feedback from L5 that sharpens L2/3 sound-evoked responses in both frequency and time domains. See Supplementary Data 1 for additional statistics. Source data are provided as a Source Data file.

cell subpopulations and found an unexpected suppression of superficial layers by L5 activity in both awake and anesthetized animals. Our results, therefore, reveal a key principle of cortical circuit operation; translaminar circuitry allows the output layer of the cortex to provide negative feedback to its upstream target,

which constantly sparsens the activity of neurons in superficial layers and sharpens their tuning in sensory space.

Our results identified L5-mediated suppression of superficial layers via both intracortical and subcortical pathways. Three lines of evidence support the substantial role of the intracortical

translaminar circuit: first, the effects of L5 manipulations were consistently the largest in L2/3, which were followed by L4 and MGv. Second, despite our likely overestimation of the onset latency for LED-triggered suppression, the latency for L2/3 suppression was no longer than that of MGv. Lastly, manipulations of IT neurons had more robust effects on L2/3 than the same manipulations of PT neurons.

Although the exact synaptic mechanisms underlying intracortical suppression remain to be identified, our data show that L5 activation preferentially recruits FS neurons over RS neurons by threefold in superficial layers. This result is consistent with L5 pyramidal cell projections to L2/3 FS neurons and small basket-like adapting interneurons in vitro[26,44]. It may appear at odds that both RS and FS units show suppression at the steady state during L5 activation. However, this co-modulation of excitatory and inhibitory neurons is explained by the recurrent network dynamics in an inhibition-stabilized network[4,8,47], as experimentally demonstrated in multiple cortical areas in awake animals[6,7,9,10,53]. Indeed, the kinetics of transiently activated L2/3 FS units is consistent with this model and supports the role of L5 excitatory inputs onto L2/3 inhibitory neurons. Nonetheless, our data do not exclude the contribution from other pathways involving inhibitory neurons in both deep and superficial layers. For example, L5 Martinotti cells and L6 basket cells inhibit pyramidal cells in L2/3[33,54–56], and somatostatin-expressing L5 non-Martinotti cells target L4 for inhibition[57,58]. Furthermore, since L2/3 somatostatin-expressing neurons play critical roles in regulating cortical tuning[6,9,59–63], their recruitment may also contribute to the L5-mediated inhibitory feedback. The existence of multiple inhibitory pathways may ensure the robust feedback suppression of superficial layers in the face of various patterns of L5 activities.

Historically, excitatory feedback projections between L5 and L2/3 pyramidal cells have been reported by numerous studies[2,24–30]. However, their estimated connectivity is relatively low, which likely accounts for the lack of observed L5 → L2/3 connectivity in other studies[33,64]. Moreover, L5 pyramidal cells also send projections to L2/3 inhibitory neurons[2,26,33,44], making it difficult to estimate the net impact of this translaminar recurrence in vivo. In our L5-selective optogenetic activation, we observed a transient or sustained increase of firing in 11% of RS units in superficial layers, in agreement with sparse excitatory connectivity of the ascending projection. Importantly, however, the fraction of activated FS units was three times higher (39%) in superficial layers, explaining the net inhibitory impact of L5 activation on L2/3 firing in vivo. Although optogenetic activation experiments need interpretation with caution due to potentially non-physiological activity patterns, our eNpHR3-mediated loss-of-function experiments provide strong evidence that L5 suppresses L2/3 activity in physiological conditions.

Our results are consistent with a recent study that reported poor recruitment of L2/3 neurons by sparse L5 activation[32]. We conceptually extend this finding by further demonstrating the net suppression of L2/3 firing, which was not observed in the previous study due to their use of calcium imaging. Surprisingly, one previous study reported that activation of Rbp4 + L5 pyramidal cells depolarized L2/3 neurons and triggered transitions to the global up state in anesthetized mice[31]. This work proposed that L5-mediated recurrent circuitry enhances and prolongs sustained activity within the cortical column, which is contrary to our data. Although the most obvious difference is their use of anesthesia, we found that the same anesthesia condition (2 mg/kg urethane) did not affect our results and confirmed brain state-independent suppression of superficial layers by L5 activation. One explanation is that, since the majority of their data are LFP or multi-unit recordings with electrodes placed 450–700 μm deep, they may reflect the measurement of optogenetically modulated L5 activity

rather than global recurrent activity (see Fig. 1b for distribution of Rbp4+ neurons across A1 cortical depth). Alternatively, since the previous work was conducted in unspecified neocortical areas, potential differences in the circuit wiring between cortical areas could account for the discrepancy. Indeed, previous studies also found opposing roles for L6 corticothalamic neurons between A1 and V1; optogenetic activation of Ntsr1 + L6 neurons suppressed the activity of superficial layers in V1[42,54,65], whereas the same manipulation in A1 enhanced it[66]. Altogether, these works may indicate that distinct layers across cortical areas provide negative feedback onto superficial layers: L5 in A1 and L6 in V1. Regardless of whether this specific scheme is correct or not, the similar negative feedback circuits found across sensory modalities highlight the importance of feedback suppression from output layers in the cortex.

L5 is known as a layer with the highest spontaneous activity[67,68]. This high-level basal firing indicates that L5 constantly sparsens and sharpens the activity of superficial layers even in quiescence, which is supported by our optogenetic inactivation data. Given the extensively associative network between L2/3 neurons, sparsening their activity may improve the information storage capacity of the cortical circuit[69,70]. It is noteworthy that, unlike the L5 activation that suppresses L2/3 via both intracortical and subcortical projections, L5 inactivation effects are found to be mediated exclusively by the intracortical pathway. A similar asymmetry between activation and inactivation was also reported in the manipulation of cortical projections onto the inferior colliculus[45]. Thus, the local intracortical loop likely allows for continuous feedback suppression in the basal state, while the more energy-consuming subcortical feedback loop takes effect only during raised activity to ensure sufficient sharpening of cortical activity. In addition, since we used pure tones with relatively high intensity (70 dB SPL) throughout our experiments, L5 recurrent connections may show different contributions to the processing of near-threshold sound stimuli. In the future, it would be interesting to investigate how the recruitment of intracortical and subcortical feedback depends on sound stimulus features, such as the intensity and spectro-temporal structures.

Although the dominant effect of L5 activation is to suppress superficial layers, we also observed transient activation in a small fraction of L2/3 RS neurons. Therefore, our results do not rule out the possibility that phasic activation of L5 neurons may regulate sensory tuning of a sparse population of L2/3 neurons by both excitatory and inhibitory mechanisms. Interestingly, ascending projections from L5 neurons are reported to lack a preference for functionally connected co-tuned subnetworks, which contrasts with L4 → L2/3 or L2/3 → L5 projections[24,32]. Thus, future work to identify the rules governing L5 → L2/3 connectivity will be critical in understanding how both excitatory and inhibitory feedbacks contribute to cortical sensory processing.

L5 neurons are known to receive most of their excitatory inputs from L2/3[2], and this descending excitatory drive has been experimentally supported both in vitro and in vivo[32,71]. These observations place the L5 → L2/3 translaminar inhibitory circuit well poised to function as a negative feedback mechanism that sparsens and stabilizes cortical activity. In contrast, recent studies reported that deep cortical layers also receive direct thalamo-cortical inputs and are able to maintain their activity in the absence of descending inputs from L2/3 neurons[14,72,73]. This finding raises the possibility that L5 could regulate superficial layers independently from its inputs from L2/3, potentially expanding the computational capacity of this translaminar circuit. Convergence of ascending, descending, and lateral inputs onto L5[14,17] may allow flexible modulation of cortical sensory processing based on various factors, such as sensory context, attention, and experience.

## Methods

**Animals.** Mice were at least 6 weeks old at the time of experiments. Tg(Rbp4-Cre)KL100Gsat/Mmucd (Rbp4-Cre; MMRRC 037128-UCD) and Tg(Tlx3-Cre)PL56Gsat/Mmucd (Tlx3-Cre; MMRRC 041158-UCD) were acquired from MMRRC. Gt(ROSA)26Sor$^{tm9(CAG-tdTomato)Hze}$/J (Ai9, JAX 007909), C57BL/6J (JAX 000664), and CBA/J (JAX 000651) were acquired from the Jackson Laboratory. Both female and male animals were used and housed at 21 °C and 40% humidity in a reverse light cycle (12 h–12 h). All experiments were performed during their dark cycle. All procedures were approved and conducted in accordance with the Institutional Animal Care and Use Committee at the University of North Carolina at Chapel Hill, as well as guidelines of the National Institute of Health.

**Viruses.** AAV8.2.hEF1a.DIO.synaptophysin.EYFP (AAV-RN2; $2.5 \times 10^{12}$ genome copies (GC)/mL) and AAV8.2.hEF1a.DIO.synaptophysin.mCherry (AAV-RN1; $2.5 \times 10^{12}$ GC/mL) were obtained from Gene Delivery Technology Core, Massachusetts General Hospital. AAV9.hsyn.Flex.ChrimsonR.tdTomato (AV6556C; $2.5 \times 10^{12}$ GC/mL) was obtained from the University of North Carolina Vector Core. AAV9.EF1a.DIO.eNpHR3.0.EYFP.WRE.hGH (v26966; $2.2 \times 10^{13}$ GC/mL) was obtained from Addgene. CAV-2.Cre ($4.0 \times 10^{11}$ GC/mL) was obtained from Plateforme de Vectorologie de Montpellier.

**Sound stimulus.** Auditory stimuli were calculated in Matlab (Mathworks) at a sample rate of 192 kHz and delivered via a free-field electrostatic speaker (ES1; Tucker-Davis Technologies). Speakers were calibrated over a range of 2–64 kHz to give a flat response (±1 dB). Click sounds were generated as 0.1-ms monopolar rectangular pulses and presented 200 trials at a 0.5-s interval. Tonal receptive fields were determined by presenting pure tones at nine frequencies (4–64 kHz, log-spaced), 70 dB SPL, 200-ms duration, nine trials at 3–4-s intervals. For testing weak LED intensities (Supplementary Fig. 2), smaller trial numbers (1–3 trials per frequency) were used. Tone stimuli were presented in semi-randomized order, and trials with and without LED were interleaved with each other. For areal mapping with intrinsic signal imaging, 3, 10, and 30 kHz pure tones (75 dB SPL, 1-s duration) were presented at a 30-s interval. Pure tone stimuli had 5-ms linear rise-fall at their onsets and offsets. Stimuli were delivered to the ear contralateral to the imaging or recording site. Auditory stimulus delivery was controlled by Bpod (Sanworks) running on Matlab.

**Intrinsic signal imaging.** Intrinsic signal images were acquired using a custom tandem lens macroscope (composed of Nikkor 35 mm 1:1.4 and 135 mm 1:2.8 lenses) and a 12-bit CMOS camera (DS-1A-01M30, Dalsa) placed in a sound isolation chamber (Gretch-Ken Industries). All mice were first implanted with a custom stainless-steel head-bar. Mice were anesthetized with isoflurane (0.8–2%) vaporized in oxygen (1 L/min), and kept on a feedback-controlled heating pad at 34–36 °C. The muscle overlying the right auditory cortex was removed, and a head-bar was secured on the skull using dental cement. The brain surface was imaged through the skull, which was kept transparent by saturation with phosphate-buffered saline (PBS). Mice were injected subcutaneously with chlorprothixene (1.5 mg/kg body weight) prior to imaging. Images of surface vasculature were acquired using green LED illumination (530 nm), and intrinsic signals were recorded (16 Hz) using red illumination (625 nm). Each trial consisted of 1-s baseline followed by a 1-s tone stimulus (75 dB pure tone with a frequency of 3, 10, or 30 kHz) and 30-s inter-trial interval. Images of reflectance were acquired at 717 × 717 pixels (covering 2.3 × 2.3 mm). Images during the response period (0.5–2 s from the tone onset) were averaged and divided by the average image during the baseline. Images were averaged across trials for each sound, Gaussian filtered, and thresholded for visualization. Individual auditory areas including A1, anterior auditory field (AAF), ventral auditory field (VAF), and secondary auditory cortex (A2) were identified based on their characteristic tonotopic organization.

**Histology.** To quantify somata distribution for pan-L5 and IT pyramidal cells, Rbp4-Cre×Ai9, Rbp4-Cre×CBA, Tlx3-Cre, or Tlx3-Cre×Ai9 mice were injected with AAV8.2.hEF1a.DIO.synaptophysin.EYFP or AAV8.2.hEF1a.DIO.synaptophysin.mCherry (30–50 nL) into L5 (400–500 µm deep) of one site within the right A1, guided by intrinsic signal imaging. To quantify somata distribution for PT pyramidal cells, C57BL/6 J or Ai9 mice were injected with CAV-2.Cre (100 nL/site) into three sites within the external cortex of the right inferior colliculus, which was visualized through the thinned skull (one medial site, 400 µm deep from the pial surface, and two lateral sites, 350 and 920 µm deep). In the same mice, AAV8.2.hEF1a.DIO.synaptophysin.EYFP (30–50 nL) was injected into L5 of one site within the right A1, guided by intrinsic signal imaging. Three weeks (pan-L5 and IT) or five weeks (PT) after injection, mice were deeply anesthetized with isoflurane and transcardially perfused first with PBS and then with 4% paraformaldehyde in PBS. The brains were removed, postfixed with 4% paraformaldehyde, and immersed in 30% sucrose in PBS for cryoprotection. The brains were coronally sectioned in 40-µm thickness with a freezing microtome, and the sections were counterstained with DAPI. Fluorescence images were acquired with a confocal microscope (Olympus FV3000RS) in 1.5- or 2-µm Z-stacks. For quantification of the signal distribution across the cortical column, images were Z-projected with the maximum intensity,

averaged across the tangential axis, and normalized by the sum of intensity across depths. The cortical depth was standardized with 0 for the brain surface and 1 for the white matter. Brains used for unit recordings were also processed and sectioned in the same way, except they did not go through perfusion. Linear probes were painted with DiI or DiO dissolved in ethanol before insertion into the brains to mark the tracks of probe penetrations. Fluorescence images of ChrimsonR.tdTomato, eNpHR3.0.EYFP, DiO, and DiI were acquired by an epifluorescence microscope (Nikon Eclipse E800). Histology figure panels were generated by overlaying signals from multiple colors using Fiji software (https://imagej.net/Fiji).

**Unit recording with optogenetic manipulation.** For pan-L5 and IT neuron activation, Rbp4-Cre×Ai9, Rbp4-Cre×CBA, or Tlx3-Cre mice were injected with AAV9.hsyn.Flex.ChrimsonR.tdTomato into L5 (400–600 µm deep from the pial surface, 120 nL/site) at two locations within the right A1, guided by intrinsic signal imaging. For PT neuron manipulations, we chose retrograde viral strategy over transgenic mice since we observed almost no recombination in A1 when we used Tg(Sim1-Cre)KJ18Gsat mice, which were reported to express Cre selectively in L5 PT neurons of the motor cortex[74]. C57BL/6 J or Ai9 mice were injected with CAV-2.Cre (100 nL/site) into three sites within the external cortex of the right inferior colliculus, which was visualized through the thinned skull (one medial site, 400 µm deep from the pial surface, and two lateral sites, 350 and 920 µm deep). In the same mice, AAV9.hsyn.Flex.ChrimsonR.tdTomato was injected into L5 at two locations within the right A1. Recordings were conducted two weeks (pan-L5 and IT) or four weeks (PT) after virus injections. It was reported that CAV-2.Cre injection in the inferior colliculus labels A1 L5 PT neurons as well as a sparse population of L6 neurons close to the white matter[16]. We avoided labeling deep L6 neurons by injecting Cre-dependent AAVs at a shallower depth (400 µm) in A1 and histologically confirmed the lack of ChrimsonR.tdTomato signal in L6 after recording in each mouse. For pan-L5 and IT neuron inactivation, AAV9.EF1a.DIO.eNpHR3.0.EYFP.WRE.hGH (280 nL) was injected into the right auditory cortex of newborn mice (postnatal day 1–3; Rbp4-Cre×Ai9, Rbp4-Cre×CBA, or Tlx3-Cre) under hypothermia anesthesia, and recordings were conducted after the mice reached six weeks old. For PT neuron inactivation, AAV9.EF1a.DIO.eNpHR3.0.EYFP.WRE.hGH (280 nL) was injected into the right auditory cortex of newborn C57BL/6 J or Ai9 mice (postnatal day 1–3). After the mice reached six weeks old, CAV-2.Cre (100 nL/site) was injected into three sites within the external cortex of the right inferior colliculus, and recordings were conducted four weeks after virus injections. In all experiments, black cement was used during head cap implantation, and silicone was placed over the exposed skull to prevent light exposure to the auditory cortex between virus injection and recording.

On the day of recording, following a small craniotomy and durotomy in the right A1 identified by intrinsic signal imaging, mice were head-fixed in the awake state, and a 64-channel silicon probe (ASSY-77-H3, sharpened, Cambridge Neurotech) was slowly (approximately 1 µm per second) inserted perpendicular to the brain surface. Spikes were monitored during probe insertion, and the probe was advanced until its tip reached the white matter, where no spikes were observed. The reference electrode was placed at the dura above the visual cortex. The probe was allowed to settle for at least 1 hour before collecting data. Unit activity was amplified, digitized (RHD2164, Intan Technologies), and acquired at 20 kHz with OpenEphys system (https://open-ephys.org). A fiber-coupled LED (ChrimsonR: 625 nm; eNpHR3.0: 595 nm) was positioned 1–2 mm above the thinned skull and a small craniotomy. In interleaved trials, LED illumination was delivered that lasted 1 s before and after tone stimuli. LED intensity was 0.5–5 mW/mm$^2$ at the surface of the brain, except for Supplementary Fig. 2c–f, where weaker intensities were used. For photoactivation with ChrimsonR, illumination with linear intensity ramp at the onset (0.3 ms) was used to minimize activation of fibers of passage[71], except for Fig. 5h and Supplementary Fig. 2a, b. For photoinactivation with eNpHR3.0, constant illumination without ramp was used. In Rbp4-Cre mice, 36.0 ± 3.3% of L5 multiunits showed photoactivation in ChrimsonR experiments, and 39.0 ± 3.9% showed photoinactivation in eNpHR3 experiments. During recording, mice sat quietly (with occasional bouts of whisking and grooming) in a loosely fitted plastic tube within a custom-built sound-attenuating enclosure. The tube was lined with fleece fabric for comfort and attenuation of noise caused by scratching. Ambient light was present in the recording chamber to prevent startling the mice when the LED was turned on. In some mice, A1 recordings were conducted in both awake and anesthetized states successively. After recording in the awake state, mice were injected with urethane (1.5–2.0 g/kg body weight) subcutaneously and kept on a feedback-controlled heating pad at 37 °C. Up and down states in A1 spikes were observed 5–10 min after injection, indicating deep anesthesia. Data collection was started at least 30 min after the urethane injection. In other mice, recordings were conducted successively in A1 and MGv. By inserting a linear probe deeper after A1 recording, we were able to reach MGv, where we observed time-locked click-sound responses. Data was collected at least 1 hr after insertion was completed. The recording locations in MGv were confirmed by both short-latency click responses and post hoc identification of probe tracks in the brain sections counterstained with DAPI. In inferior colliculus recording experiments, a head cap was implanted the day before recording without covering the right auditory cortex and the inferior colliculus. The skull was thinned until the inferior colliculus was visible through the skull, and both the auditory cortex and inferior colliculus were covered with silicone. On the day of recording, a small craniotomy and durotomy were made in

the inferior colliculus, and a 64-channel silicon probe was inserted down to around 1 mm from the surface. The channels corresponding to the external cortex and central nucleus of the inferior colliculus were determined by both short-latency click responses and post hoc identification of probe tracks in the brain sections. In some mice, recordings were made successively in two locations of the inferior colliculus. A fiber-coupled LED (ChrimsonR: 625 nm) was positioned 1–2 mm above the skull over the auditory cortex.

**Analysis of unit recording data**. Single- and multi-units were isolated using Kilosort or Kirosort2 software (https://github.com/cortex-lab/KiloSort, https://github.com/jamesjun/Kilosort2) and spike-sorting graphical user interface (Phy; https://github.com/cortex-lab/phy). Single-unit isolation was confirmed based on the inter-spike interval histogram (ISI violation; less than 5% of the spikes in the refractory period, which was 2 ms for A1 RS units and 1.5 ms for A1 FS units and MGv units, after correction for the overall spike frequency) and the consistency of the spike waveform. The number of putative ChrimsonR-expressing single-units was likely underestimated since many photoactivated L5 units exceeded the ISI violation threshold and were classified as multi-units. Multi-unit spikes were calculated by combining all the spikes within each depth bin (Fig. 1g, k, Fig. 2g, j, Fig. 6e, i, an Fig. 7a, e, and Supplementary Fig. 7b, d), layer (Fig. 2b–e, i, l, Fig. 4, Fig. 5d–g, and Supplementary Fig. 2c–f), entire MGv (Fig. 5d–g), IC subdivision (Supplementary Fig. 5c), or layer across mice (Fig. 5h). Fast-spiking units were identified based on their small trough-peak interval (≤0.4 ms). Positions of cortical surface, layers, and white matter were identified by current source density analysis and the distribution of multi-unit spikes. Layer 4 was identified as the early sink in the current source density analysis. Channels between the L4 lower border and the white matter were equally divided into L5 and L6. For single-unit analyses, only units within the lowest third of L5–L6 were included in L6 to avoid contamination of L5 units expressing ChrimsonR or eNpHR3.0. Putative directly-photoactivated units invading the superficial layers (1 unit in Fig. 2b, c, 2 units in Fig. 2h, i, and 2 units in Fig. 5d, e) were included in single-unit analyses but were excluded from multi-unit analyses to avoid their dominance of the total spike count. For visualization of the laminar profile of L5 manipulation effects, cortical depth was morphed into a normalized cortical column in each mouse such that L1–3, L4, L5, and L6 are represented by 4, 4, 6, and 6 bins, respectively. For FS units, the bin numbers were 2, 2, 3, and 3, respectively, to account for their sparsity. PSTHs were generated at 25 ms-bin, except for Fig. 5h, where 0.5 ms-bin was used, and Fig. 5d, f, Fig. 7c, g, Supplementary Fig. 2a, and Supplementary Fig. 5c, where 10 ms-bin was used to visualize events with faster kinetics. In Fig. 5h, PSTHs were smoothened by a 2-ms Gaussian kernel to facilitate fitting.

Effects of optogenetic manipulations were quantified as modulation index (MI), which was calculated as $(L - C)/(L + C)$, where L represents the activity in LED trials and C represents the activity during No LED control trials. Thus, MI ranges from −1 to 1, where a value of −1 represents a complete loss of activity, 1 represents the emergence of activity from nothing, and 0 represents no change. This index is advantageous in describing population dynamics over the simple ratio (L/C), which gives an extremely wide range of values from zero to infinity. MI was calculated for either spontaneous firing rate or tone-evoked spikes. Spontaneous firing rate was quantified in the 500-ms window preceding tones. Tone-evoked responses were quantified after subtraction of the baseline firing rate 0–200 ms before tone onsets and calculated as the sum of positive-going PSTH during the sound duration. Thus, tone-evoked responses do not include the increase in the spontaneous firing rate caused by LED. Unit-frequency pairs were judged as significantly responsive if they fulfilled two criteria: (1) PSTH had to exceed a fixed threshold value at the same time bin in more than one third of trials. (2) Trial-averaged PSTH had to exceed a fixed threshold value. Threshold for excitation (3.7× standard deviation during baseline period before LED) was determined by ROC analysis to yield a 90% true positive rate in tone-evoked responses. For spontaneous activity, single-units with MI higher than 1/9 (corresponding to 25% increase in spike count) were classified as enhanced, and units with MI lower than −1/9 (corresponding to 20% decrease) were classified as suppressed. To detect transient firing changes with confidence, single-units with spontaneous firing rate below 0.4 Hz in both No LED and LED conditions were classified as unchanged in Fig. 7d, h. For tone-evoked activity, single-units with MI higher than 1/6 (corresponding to 40% increase in spike count) were classified as enhanced, and units with MI lower than −1/6 (corresponding to 29% decrease) were classified as suppressed. Higher thresholds were used for tone-evoked activity due to its trial variability. For classifying divisive and subtractive transformations, single-units were fit by threshold-linear model, in which only the data points with non-zero responses (over 10% of the peak response amplitude) in both LED and No LED conditions and the first data point below the threshold were used for linear fitting. Single-units that were fit with Pearson's correlation coefficient (R) > 0.4 are included as good-fit units. Units with slopes ≥ 1.3, slopes ≤ 0.7, normalized y-intercepts ≥ 0.15, and normalized y-intercepts ≤ −0.08 were classified as multiplicative, divisive, additive, and subtractive, respectively. Tone response latency was quantified as the time when the spike rate exceeded 20% of the maximum in the baseline-subtracted PSTH. Decay time was quantified as the time from 100% to 30% maximum. Full-width of half-maximum (FWHM) was the time between 50% maximum points in the rising and falling phases. We applied the following protocol to fit the suppression of activity in individual layers with

exponential curves. (i) We used the PSTH of multi-unit spikes in each layer combined across all mice and normalized to the baseline level before LED. (ii) Units with obvious LED-triggered increases in spikes were excluded. (iii) PSTH between suppression delay (see below) and 60 ms after LED onset was used for fitting. (iv) Suppression delay (time between LED onset to the start of the exponential curve) was determined by testing delay values at a 1-ms increment and finding the value that generated the best fit. The exact delay values in Fig. 5h were determined as the time when the fit curve fell below one. (v) PSTH was smoothened with a Gaussian-weighted moving average over 2-ms window for denoising. (vi) Fitting was conducted without forcing the starting level or the floor level. (vii) For L2/3, PSTH was fit with a sum of two exponentials. The delay of the second exponential was also automatically selected. L4 and MGv data were fit with a single exponential since fitting with two exponentials returned negligible contributions of the second exponential (L4: 0, MGv: 0.03).

**Pupillometry**. The eye contralateral to the recording site was imaged with an infrared camera (Teledyne Dalsa, Genie Nano M640-NIR) equipped with a zoom lens (FUJINON HF50HA-1B with Computar VM100 Extention Tube Kit) placed 10 cm from the eye. An infrared light source was positioned to illuminate the eye. Additional light sources with red and/or amber LEDs were positioned to provide low-intensity illumination such that the mouse's pupil was approximately mid-range in diameter. Pupil movie was acquired and synchronized with sound presentation trials using Bonsai software (https://bonsai-rx.org/). Images (480 × 640 pixels, downsampled to 120 × 160 pixels) were collected at 30 Hz. The diameter of the pupil was measured using a custom-written program in Matlab. First, an intensity threshold was determined for each movie, and the images were binarized to extract the pupil pixels. Binarized images were processed with opening, closing, and filling operations to smoothen the boundary of the pupil pixels. Pupil diameter was determined as the max Feret diameter of the extracted pupil pixels. Following the automated calculation of the pupil diameter, image frames with undetectable pupils (such as the frames with the pupil hidden behind the eyelid) were manually excluded with the help of a custom-written deviant-detection program. Pupil size was normalized to the minimum and maximum diameters throughout the recording in each mouse. Trials were classified into four bins based on the normalized diameter during the baseline window 0–300 ms preceding the LED onset. For No LED control trials, a corresponding baseline window within the trial was used.

**Statistical analysis**. All data are presented as mean ± SEM or median with 25th and 75th percentiles, as stated in the figure legend. Statistically significant differences between conditions were determined using standard two-sided parametric or nonparametric tests in Matlab. Wilcoxon signed-rank test was used for one-sample nonparametric tests. Paired t-test was used for paired data, and Wilcoxon rank-sum test was used for independent group comparisons. Bonferroni correction was used for multiple comparisons, and corrected p values were reported. In cases where parametric statistics are reported, the normality of data distribution was tested with one-sample Kolmogorov-Smirnov test. Randomization is not relevant for this study because there were no animal treatment groups. All "n" values refer to the number of single-units except when explicitly stated that the n is referring to the number of mice, number of unit-frequency pairs, or number of sections. Experiments were not performed blind. Sample sizes were not predetermined by statistical methods but were based on those commonly used in the field.

**Reporting summary**. Further information on research design is available in the Nature Research Reporting Summary linked to this article.

## Data availability

Source data for all figures are provided with this paper as a supplementary data file. Other data that support the findings of this study are available from the corresponding authors upon reasonable request.

## Code availability

Custom Matlab codes used in this study will be made available from the corresponding author upon reasonable request.

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

## Acknowledgements

We thank Paul Manis, Jose Rodriguez-Romaguera, and the members of the Kato Lab and the Rodriguez-Romaguera Lab for their advice throughout the project and comments on the manuscript. We thank Gonçalo Lopes for his support in setting up Bonsai for pupil monitoring. This work was supported by NIDCD (R01DC017516), Pew Biomedical Scholarship, Whitehall Foundation, Klingenstein-Simons Fellowship, Foundation of Hope (H.K.K.), Toyobo Biotechnology Foundation, and Japan Society for the Promotion of Science Overseas Research Fellowship (K.O.). Microscopy was performed at the UNC Neuroscience Microscopy Core (RRID: SCR_019060), supported, in part, by funding from the NIH-NINDS Neuroscience Center Support Grant P30 NS045892 and the NIH-NICHD Intellectual and Developmental Disabilities Research Center Support Grant U54 HD079124.

## Author contributions

K.O. conducted all the experiments. K.O. and H.K.K. designed the project, analyzed data, and wrote the manuscript.

## Competing interests

The authors declare no competing interests.
