## [Peer Review File · Nature Communications]

Translaminar Recurrence from Layer 5 Suppresses Superficial Cortical LayersREVIEWER COMMENTS

Reviewer #1 (Remarks to the Author):

In this study, Onodera and Kato study recurrent excitatory projections from layer 5 to superficial L2/3 and found that in addition to the classical excitatory effect there is a clear inhibitory effect that is also mediated by the action of the thalamocortical projections.

This is a comprehensive study and the data quite robust and convincing. Authors have made elegant experiments combining activating and deactivating of different cell and pathways, not only involving intracortical, but also including the thalamocortical pathway contribution to this effect.

Overall, the manuscript is well written. Nevertheless, I have some comments and suggestion that hopefully will help author to improve the final version.

A very important finding in my view is the fact that results in anesthetized and awake is virtually very similar. This is of paramount importance as it is always a big issue nowadays when we get papers that many reviewers/editors and journal basically neglect studies based solely on anesthetized preparations. This study is extremely important in that it used both preparations and that results are generally alike. This should be further emphasized in the title and abstract with a sentence similar to what authors state on page 7: 'These results demonstrate that the suppressive effect of L5 onto superficial layers is robust regardless of the arousal level'

I perfectly understand the magnitude of the work and that there should be a lot of figures, but I find the figures difficult to digest. They are too informative and too dense and I urge authors to somehow simplify. I know this is challenging, but I find panels too small and difficult to appreciate details. I don't think there is a figure limitation number in Nat Comm so perhaps it would help to break some of the figures into two or even three.... For example, figs 1, 3 and 6 are made of almost 20-25 panels. Also, the colors used for the dot rasters and PSTH in the control and LED are not so appealing and clear to distinguish in my view. Scatterplots like those on fig 1i or 1m might be better visualized using violin plots, which are more informative and clearer.

An important conceptual issue is related to the use of the term 'tuning curves'. What authors refer to as tuning curves are not really such. What authors have recorded are the so-called 'isointensity functions'. Tuning curves are constructed from different intensities and frequencies, that way you can really calculate the BF and threshold etc.... Moreover, if I understood correctly much of their recordings have

been made at high intensities, around 70 dB-SPL. So, this should be clearly stated, perhaps recurrent connections may play a different functional role at level stimulation close to threshold.

On page authors refer the reader to the methods section for details on MI computation etc. I think it will be good to get a small explanation here of what is it and the rationale to use this matrix or index. Having 1-2 lines here explaining what is MI and how you calculated will be useful for the general readership. I also wonder if author might consider using some sort of bootstrapping analysis for significant effects between control and LED rather than the MI or in addition to.

It will be also important to mention, at beginning or results (and may be invent good to have a summary Table) of how many single- and multinet have been recorded in the whole ms and different experimental manipulations. It is somehow hidden in the figs and not totally clear.

Where in the AC field recordings were made. A1? AAF ? please specify.. did you pool all together?

Why normality was not tested? I think tis should be done---In cases where parametric statistics are reported, data distribution was assumed to be normal, but this was not formally tested.

Reviewer #2 (Remarks to the Author):

REVIEWER COMMENTS

Reviewer #1 (Remarks to the Author):

In the manuscript entitled "Translaminar Recurrence from Layer 5 Suppresses Superficial Cortical Layers", Onodera and Kato combined optogenetic manipulations of L5 neurons with single-unit recordings in the A1, and found that layer 5 activation suppressed the activity of superficial layers and also sharpened tone-evoked responses of superficial layers. In addition, by manipulation of IT neurons or PT neurons, they further suggested that this suppressive effect was mainly mediated by intracortical IT neurons. Finally, by analyzing fast spiking units, they suggested that layer 5 IT neurons projected preferentially to fast spiking inhibitory neurons in the superficial layers. This paper is well written and

the experimental results are easy to follow. I have a few points of concern and some suggestions for possible improvements of their presentation.

Major points

The claim "our findings challenge the classical view of feedforward cortical circuitry..." is too strong and needs to be tuned down. L5 excitatory neurons project to L2/3 pyramidal neurons and different types of inhibitory neurons, and therefore the net effect on L2/3 by activating L5 excitatory neurons is not that simple. This could depend on many factors, including brain states (tested by the authors), stimulus features and strengths, local microcircuit organization, and the subtype of L5 neurons that was activated. Artificially activating Rbp4-positive L5 neurons or completely silencing these neurons by optogenetics does not necessarily mimic the endogenous L5 activity in certain conditions. Therefore, their results only added a possible circuit but not challenge the classical view.

Another major point is why the authors only looked at fast-spiking PV interneurons, as in their recent study, they suggested a strong contribution to cortical tuning by somatostatin-expressing (SOM) interneurons in A1. Do also SOM neurons play a role in L5-mediated suppression in superficial layers?

Minor concerns:

- 1, Providing more information of Rbp4-labeled L5 neurons would be helpful. What is the fraction of L5 neuron labeled? Are there any L5 interneurons?
- 2, Fig1f and g, why did they use 4 kHz tone results for photoactivation experiments, but use 16 kHz tone for photoinhibition experiments? Any specific reason, or just choose the best results to show?
- 3, The test of anesthesia in Fig2, why did they test urethane but not isoflurane?
- 4, What was the background noise sound level in their setup? What are the components of the noise?
- 5, The black traces in Fig4a and e look quite different. If I understand correctly, they should be largely the same?
- 6, Labeling PT neurons through injection of CAV2-cre into the inferior colliculus needs to be verified. Any labeling in other layers, like layer 6? Do these neurons overlap with Rbp4-positive neurons?
- 7, Why did PT neuron activation only suppress spontaneous but not tone-evoked firing?
- 8, Do other cortical areas show the same suppressive effect, e.g. V1?
- 9, Language needs to be carefully read by a native speaker.

Response to Reviewers:

We are delighted that all two reviewers appreciated our original submission and found that *“This is a comprehensive study and the data quite robust and convincing. Authors have made elegant experiments combining activating and deactivating of different cell and pathways”* and *“This paper is well written and the experimental results are easy to follow.”* We appreciate all the constructive suggestions that helped us improve our manuscript. We have performed additional statistical analyses and revised the figures and text as suggested by the reviewers. Below, please find our response to all of the specific comments made by the reviewers. Text changes are highlighted in red in the revised manuscript.

Reviewer 1:

In this study, Onodera and Kato study recurrent excitatory projections from layer 5 to superficial L2/3 and found that in addition to the classical excitatory effect there is a clear inhibitory effect that is also mediated by the action of the thalamocortical projections.

This is a comprehensive study and the data quite robust and convincing. Authors have made elegant experiments combining activating and deactivating of different cell and pathways, not only involving intracortical, but also including the thalamocortical pathway contribution to this effect.

Overall, the manuscript is well written. Nevertheless, I have some comments and suggestion that hopefully will help author to improve the final version.

We thank Reviewer 1 for their kind words regarding our manuscript and the constructive comments for improvements. We are especially delighted that this reviewer found our work ‘a *comprehensive study and the data quite robust and convincing.*’ Below are the responses to this reviewer’s comments.

Major Comments

A very important finding in my view is the fact that results in anesthetized and awake is virtually very similar. This is of paramount importance as it is always a big issue nowadays when we get papers that many reviewers/editors and journal basically neglect studies based solely on anesthetized preparations. So this study is extremely important in that used both preparations and that results are generally alike. This should be further emphasized in the title and abstract with a sentence similar to what authors state on page 7: ‘These results demonstrate that the suppressive effect of L5 onto superficial layers is robust regardless of the arousal level.’

We are pleased that the reviewer appreciated the importance of our results being robust regardless of the arousal level. We agree that our data suggests the brain state-invariant principle of the L5-L2/3 translaminal feedback modulation, and this point should be more emphasized in the manuscript. We hesitate to change the title since we think the current title already conveys the general principle in a concise sentence, and adding words may rather give the impression of a narrow focus of the work. Instead, we updated the Summary to state:

“Here, we use layer-selective optogenetic manipulations in the primary auditory cortex to demonstrate that feedback inputs from L5 suppress the activity of superficial layers regardless of the arousal level, contrary to the prediction from their excitatory connectivity.”

We also updated a sentence in the first paragraph of the Discussion to state:

“In this study, we took advantage of genetic and viral tools to selectively manipulate L5 pyramidal cell subpopulations and found an unexpected suppression of superficial layers by L5 activity in both awake and anesthetized animals.”

I perfectly understand that magnitude of the work and that there should be a lot of figures, but I find the figures difficult to digest. They are too informative and too dense and I urge authors to somehow simplify. I know this is challenging, but I find panels too small and difficult to appreciate details. I don't think there is a figure limitation number in Nat Comm so perhaps it would help to break some of the figures into two or even 3.... For example, figs 1, 3 and 6 are made of almost 20-25 panels. Also, the colors used for the dots rasters and PSTH in the control and LED are not so appealing and clear to distinguish in my view. Scatterplots like those on fig 1i or 1m might be better visualized using violin plots, which are more informative and clearer.

We thank the reviewer for these suggestions for improving the data presentation. We agree that our figures were overall dense and that some simplification would benefit visualization. To address this, we first removed all the pie charts showing the fraction of enhanced and suppressed units (original Fig. 1i, 1m, 2h, 2k, 3c, 3k, 6f, 6g, 6j, and 6k) since these small panels showed redundant information with the scatter plots next to them. We now moved all these pie charts to new Supplementary Figures 1, 3, and 8. We also removed the original Figures 3d and 3l, which showed BF invariance with optogenetic manipulation, and moved them to a new Supplementary Figure 3. We also enlarged the characters and panels wherever appropriate and made the orange color darker in the raster plots to improve visibility. We decided to stick to the scatter plots in Fig. 1i or 1m since several other figures with a similar data type (e.g., scatter plots in Fig. 2 and Fig. 6) had relatively small sample sizes, and the journal encourages the authors to display individual data points. We nevertheless liked the idea of violin plots and therefore used them in the updated Fig. 3d and 3k, which did not show the data distribution in our original manuscript. We hope these changes made the figures easier to understand.

An important conceptual issue is related to the use of the term ‘tuning curves’. What authors refer to as tuning curves are not really such. What authors have recorded are the so-called ‘isointensity functions’. Tuning curves are constructed from different intensities and frequencies, that way you can really calculate the BF and threshold etc.... Moreover, if I understood correctly much of their recordings have been made at high intensities, around 70 dB-SPL. So, this should be clearly stated, perhaps recurrent connections may play a different functional role at level stimulation close to threshold.

We may be confused about this, but our understanding is that ‘tonal receptive fields’ are constructed from various tone frequencies and intensities, but the phrase ‘tuning curves’ seems to be commonly used in broader contexts. We would like to keep the term ‘tuning curves’ so that

general non-auditory readers could understand the concept, but we also inserted 'isointensity functions' to clearly define our experiments and avoid any potential confusion. The sentence that had the phrase 'tuning curve' in the original manuscript is now updated as:

"We generated tuning curves of RS single-units by presenting pure tones across a range of frequencies (isointensity functions at 70 dB SPL, 4–64 kHz, 70 dB SPL, 0.2 s)."

We also acknowledged the potential different functional contributions of L5 recurrence depending on the sound intensities in the Discussion as:

"In addition, since we used pure tones with relatively high intensity (70 dB SPL) throughout our experiments, L5 recurrent connections may show different contributions to the processing of near-threshold sound stimuli. In the future, it would be interesting to investigate how the recruitment of intracortical and subcortical feedback depends on sound stimulus features, such as the intensity and spectro-temporal structures."

On page authors refer the reader to the methods section for details on MI computation etc. I think it will be good to get a small explanation here of what is it and the rationale to use this matrix or index. Having 1-2 lines here explaining what is MI and how you calculated will be useful for the general readership. I also wonder if author might consider using some sort of bootstrapping analysis for significant effects between control and LED rather than the MI or in addition to.

We have inserted a description of MI in the Results as:

"MI was calculated as $(L-C)/(L+C)$, where L represents the activity in LED trials and C represents the activity during No LED control trials. Thus, MI ranges from -1 to 1, where a value of -1 represents a complete loss of activity, 1 represents the emergence of activity from nothing, and 0 represents no change."

In addition, we inserted the same description together with a justification in the Methods as:

"Effects of optogenetic manipulations were quantified as modulation index (MI), which was calculated as $(L-C)/(L+C)$, where L represents the activity in LED trials and C represents the activity during No LED control trials. Thus, MI ranges from -1 to 1, where a value of -1 represents a complete loss of activity, 1 represents the emergence of activity from nothing, and 0 represents no change. This index is advantageous in describing population dynamics over the simple ratio (L/C), which gives an extremely wide range of values from zero to infinity."

We also performed direct statistical comparisons of the firing rates between LED and Control conditions, and they showed consistent results as MI. (e.g., Fig. 1i, L2/3: $p = 1.16 \times 10^{-8}$; L5: $p = 7.78 \times 10^{-8}$; L6: $p = 0.133$. Fig. 1m, L2/3: $p = 2.56 \times 10^{-7}$; L5: $p = 9.89 \times 10^{-9}$; L6: $p = 0.0122$; Wilcoxon signed-rank test after Bonferroni correction for multiple comparison). We prefer to display data in MI, since showing the firing rate distributions in both LED and Control conditions for each layer would make figures even busier. Also, we did not use ratio change (L/C) since this readout shows an infinite range of values, as stated in the newly inserted sentence above.

It will be also important to mention, at beginning or results (and may be invent good to have a summary Table) of how many single- and multinet have been recorded in the whole ms and different experimental manipulations. It is somehow hidden in the figs and not totally clear.

This information was summarized as a table in Supplementary Data 1, as required by the journal. In addition, we also had the numbers of data points in both the figures and figure legends. Since the number of data points are different across figures or even across panels within a figure, we found it hard to include this information at the beginning of Results.

Where in the AC field recordings were made. A1? AAF ? please specify.. did you pool all together?

As described in the Results and Methods sections, we targeted all recordings to A1 mapped with intrinsic signal imaging (please also see Fig. 1d for a representative map).

Why normality was not tested? I think tis should be done---In cases where parametric statistics are reported, data distribution was assumed to be normal, but this was not formally tested.

We thank this reviewer for pointing out this potential issue. We have now tested normality for all the datasets that used parametric t-tests and found that the BW_{70} difference data (Fig. 3d L2/3 and Fig. 3k L2/3; in the original manuscript Fig. 3e and 3m) did not pass the one-sample Kolmogorov-Smirnov normality test. Therefore, we now revised the statistics for these two experiments to the non-parametric Wilcoxon signed-rank test with Bonferroni correction, and the results stayed identical. For consistency, other BW_{70} difference data (Supplementary Fig. 9; in the original manuscript Supplementary Fig. 6) were also revised with a non-parametric test, although they passed the normality test. All other datasets passed the normality test, so we kept the t-test statistics in our original manuscript. These data are now updated in both the main text and figure legend, and the p values for the normality test are included in Supplementary Data 1. We also revised the sentence in the Methods as:

“In cases where parametric statistics were reported, the normality of data distribution was tested with one-sample Kolmogorov-Smirnov test.”

Reviewer 2:

In the manuscript entitled "Translaminar Recurrence from Layer 5 Suppresses Superficial Cortical Layers", Onodera and Kato combined optogenetic manipulations of L5 neurons with single-unit recordings in the A1, and found that layer 5 activation suppressed the activity of superficial layers and also sharpened tone-evoked responses of superficial layers. In addition, by manipulation of IT neurons or PT neurons, they further suggested that this suppressive effect was mainly mediated by intracortical IT neurons. Finally, by analyzing fast spiking units, they suggested that layer 5 IT neurons projected preferentially to fast spiking inhibitory neurons in the superficial layers. This paper is well written and the experimental results are easy to follow. I

have a few points of concern and some suggestions for possible improvements of their presentation.

We thank Reviewer 2 for their helpful suggestions for improving our manuscript, and we are pleased that this reviewer thought that “*this paper is well written and the experimental results are easy to follow.*” Please see our responses to this reviewer below.

Major points

The claim “our findings challenge the classical view of feedforward cortical circuitry...” is too strong and needs to be tuned down. L5 excitatory neurons project to L2/3 pyramidal neurons and different types of inhibitory neurons, and therefore the net effect on L2/3 by activating L5 excitatory neurons is not that simple. This could depend on many factors, including brain states (tested by the authors), stimulus features and strengths, local microcircuit organization, and the subtype of L5 neurons that was activated. Artificially activating Rbp4-positive L5 neurons or completely silencing these neurons by optogenetics does not necessarily mimic the endogenous L5 activity in certain conditions. Therefore, their results only added a possible circuit but not challenge the classical view.

We thank this reviewer for their careful consideration of factors influencing the roles of L5 recurrent projections onto L2/3. To address this, we have updated the last sentence in the Summary as:

“Together, our findings establish a translaminar inhibitory recurrence from deep layers that sharpens feature selectivity in superficial cortical layers.”

Furthermore, to acknowledge the potentially different functional contributions of L5 recurrence depending on the stimulus features and strengths (please also see our response to Reviewer 1’s third comment), we now mention this possibility in the Discussion as:

“In addition, since we used pure tones with relatively high intensity (70 dB SPL) throughout our experiments, L5 recurrent connections may show different contributions to the processing of near-threshold sound stimuli. In the future, it would be interesting to investigate how the recruitment of intracortical and subcortical feedback depends on sound stimulus features, such as the intensity and spectro-temporal structures.”

Another major point is why the authors only looked at fast-spiking PV interneurons, as in their recent study, they suggested a strong contribution to cortical tuning by somatostatin-expressing (SOM) interneurons in A1. Do also SOM neurons play a role in L5-mediated suppression in superficial layers?

This is a very important point. We agree that it would be of great interest to investigate how other inhibitory neuron subtypes, such as SOM interneurons, contribute to the translaminar recurrent circuitry. Unfortunately, unit recording allows us to distinguish only between fast-spiking (which are mostly PV interneurons) and regular-spiking neurons. Since most SOM interneurons are regular-spiking, there is no established method to distinguish SOM interneurons from excitatory pyramidal neurons electrophysiologically. ChR-guided photo-tagging (PINP) strategy is also unfeasible since we need Cre-dependent optogenetic control for

L5 neurons. Potentially more sophisticated methods, such as SOM cell calcium imaging during simultaneous optogenetic manipulations of L5 neurons, may be able to answer this question in the future, but we currently do not have such a system established in our laboratory.

As we were aware that our data do not exclude the additional contribution of non-PV inhibitory neurons, in the original manuscript, we had a paragraph in the Discussion that described the potential roles of deep layer inhibitory neurons. However, we now realize that our previous discussion did not cover the potential contribution of SOM interneurons in the superficial layers. Therefore, we revised this paragraph to state:

“Nonetheless, our data do not exclude the contribution from other pathways involving inhibitory neurons in both deep and superficial layers. For example, L5 Martinotti cells and L6 basket cells inhibit pyramidal cells in L2/3^{33,54–56}, and somatostatin-expressing L5 non-Martinotti cells target L4 for inhibition^{57,58}. Furthermore, since L2/3 somatostatin-expressing neurons play critical roles in regulating cortical tuning^{6,9,59–63}, their recruitment may also contribute to the L5-mediated inhibitory feedback. The existence of multiple inhibitory pathways may ensure the robust feedback suppression of superficial layers in the face of various patterns of L5 activities.”

Minor concerns:

1, Providing more information of Rbp4-labeled L5 neurons would be helpful. What is the fraction of L5 neuron labeled? Are there any L5 interneurons?

In our Methods section of the original manuscript, we described that “In Rbp4-Cre mice, $36.0 \pm 3.3\%$ of L5 multiunits showed photoactivation in ChrimsonR experiments, and $39.0 \pm 3.9\%$ showed photoinactivation in eNpHR3 experiments.” Although these values may not represent the accurate percentage of Rbp4-labeled L5 neurons, they likely provide a rough estimate. A previous large-scale single-cell transcriptomics study reported that Rbp4-Cre-labeled neurons overlapped with various L5 excitatory neuron transcriptome types but not with inhibitory neuron types (Tasic et al., Nat Neurosci 2016; doi: 10.1038/nn.4216). Also, immunostaining with anti-GABA antibody did not overlap with Rbp4-Cre in another study (Beltramo et al., Nat Neurosci 2013; cited in our manuscript), further supporting the excitatory identity of Rbp4-Cre neurons. Since the former paper was not cited in our original manuscript, we now included it as:

“In this strain, transgene expression is restricted to excitatory neurons in both superficial and deep sublayers of L5^{31,35,36}.”

2, Fig1f and g, why did they use 4 kHz tone results for photoactivation experiments, but use 16 kHz tone for photoinhibition experiments? Any specific reason, or just choose the best results to show?

We played pure tones with nine frequencies (4-64 kHz, log-spaced) to all mice in our experiments, as shown more explicitly in Figure 3. Figures 1f and 1j display the PSTHs of two representative units in response to their best frequency stimuli. We now included this information in the legend as:

“gray shading, tone stimulation at the unit’s best frequency.”

3, The test of anesthesia in Fig2, why did they test urethane but not isoflurane?

There are two reasons for the choice of urethane. First, it has been reported that isoflurane anesthesia elevates the auditory brain stem response threshold by greater than 27 dB compared to ketamine anesthesia (Ruebhausen et al., Hearing Research 2012; doi: 10.1016/j.heares.2012.04.005), suggesting the negative influence of isoflurane on hearing. We use isoflurane for intrinsic signal imaging since it still leaves enough signal for mapping, but we did not want to use it for quantitative experiments. Second, since the main motivation of Figure 2 was to compare our results with the previous study (Beltramo et al., Nat Neurosci 2013), we chose the same anesthesia condition as this study.

4, What was the background noise sound level in their setup? What are the components of the noise?

Please see the figure on the right for the noise measurement at our recording rig inside the sound attenuation chamber. The small peak around 4 kHz is an electric noise from the amplifier/DAQ and not sound-related. As you see, the power of the white noise is restricted to <1 kHz (outside the mouse hearing range), and the rest is below the detection level of our measurement system.

5, The black traces in Fig4a and e look quite different. If I understand correctly, they should be largely the same?

The reason was explained in the legend of Fig. 4 in the original manuscript as:

“Frequencies with significant responses in either LED or No LED conditions were included in multi-unit spikes. Only multi-unit data with significant responses in both LED and No LED conditions were included in kinetics analyses. These necessary selection criteria biased FWHM towards larger values in L5 activation experiments since mice with small responses in No LED trials tended to be excluded due to the loss of responsiveness in No LED trials.”

6, Labeling PT neurons through injection of CAV2-cre into the inferior colliculus needs to be verified. Any labeling in other layers, like layer 6? Do these neurons overlap with Rbq4-positive neurons?

Thank you for pointing out this missing information. As shown in the cellular distribution in Fig. 6c, our injection strategy did not label layer 6 neurons. (We now moved the scale bar outside the image for clarity.) However, this is actually because we injected cre-dependent AAVs relatively shallow (400 μ m deep) in A1. In the earlier preliminary experiments injecting CAV-2.Cre into the inferior colliculus of tdTomato reporter mice, we found tdTom in L5 PT neurons as well as a sparse population of deep L6 neurons close to the white matter (consistent with a previous report: Williamson and Polley, Elife 2019, DOI: 10.7554/eLife.42974). We found that

we could avoid L6 labeling by injecting AAVs at a shallow depth, and therefore this condition was used throughout our experiments. We now included this information in the Methods as:

“It was reported that CAV-2.Cre injection in the inferior colliculus labels A1 L5 PT neurons as well as a sparse population of L6 neurons close to the white matter¹⁶. We avoided labeling deep L6 neurons by injecting Cre-dependent AAVs at a shallower depth (400 μm) in A1 and histologically confirmed the lack of ChrimsonR.tdTomato signal in L6 after recording in each mouse.”

Unfortunately, the requirement of Cre for both CAV-2.Cre and Rbp4-Cre did not allow us to examine their overlap directly. However, a previous large-scale transcriptomics study found that Rbp4-Cre-labeled neurons overlapped with all known L5 excitatory neuron transcriptome types, including both IT and PT subtypes (Tasic et al., Nat Neurosci 2016; doi: 10.1038/nn.4216), suggesting the pan-L5 nature of Rbp4-Cre. Moreover, our Fig. S6 (originally Fig. S4) shows that Rbp4-Cre population sends projections to the external cortex of the inferior colliculus. These data suggest that CAV-2.Cre-labeled neurons and Rbp4-Cre neurons overlap with each other.

7, Why did PT neuron activation only suppress spontaneous but not tone-evoked firing?

This is an interesting point, but we do not know the exact answer. Throughout our experiments, we reproducibly found that the L5 manipulation effects on L2/3 neurons were more pronounced in the spontaneous than the sound-evoked activity. This was true not just for PT neurons but also for IT and pan-L5 (Rbp4) manipulations (e.g., compare Fig. 1i,1m vs. 3c, 3j; Fig. 6f vs. 6g; 6j vs. 6k). Considering that the L2/3 spontaneous activity suppression is already smaller for PT activation compared to IT or Rbp4 activation, it may be possible that its effect on tone-evoked activity was too small to be detected.

In general, apart from the L5 manipulations, our experience is that sound-evoked activity is more resistant to optogenetic suppression than spontaneous activity. We also had a similar observation in our recent paper in which we directly inactivated A2 activity with PV cell activation (Fig. 8f of Kline et al., Nat Commun 2021, doi: 10.1038/s41467-021-24758-6). One speculation is that the non-linear input-output relationship in the cortical firing may result in the differential effects of suppressive inputs depending on the cortical activity level.

8, Do other cortical areas show the same suppressive effect, e.g. V1?

This is another interesting point. We already had a paragraph in the Discussion considering the potential differences between cortical areas. A previous study reported that Rbp4 neuron activation triggered transitions of the cortex to the global up state in anesthetized mice (Beltramo et al., Nat Neurosci 2013, doi: 10.1038/nn.3306). The experiments in this paper were conducted in the unspecified sensorimotor cortex, and therefore the effects of L5 manipulation may differ across cortical areas. However, as we discussed in the manuscript, their LFP measurement at 450-700 μm deep may reflect the direct optogenetic effects on L5 neurons rather than the global cortical activity, leaving the conclusion unclear. Although it would be of interest to examine the roles of L5 recurrence in other cortical areas, including V1, it is beyond the scope of our current manuscript.

9, Language needs to be carefully read by a native speaker.

Thank you for the suggestion. Our original manuscript was already read by several native speakers and went through a grammar checking service, but we had our updated manuscript re-read by another one. (Most of the suggested edits were addition/removal of 'the,' and we did not mark them in red.)

REVIEWER COMMENTS

Reviewer #1 (Remarks to the Author):

I would like to thank the authors for the effort and the good job , and I am generally happy with the revision, and most of my comments (although not all) have been taken care and addressed.

But the issue of tuning curves is simply incorrect. Authors try to convince the reader that tuning curves is equal to iso-intensity functions. I am sure they do this with their best intention, but in my view this is more harmful than helpful for the general audience

I don't think there is a confusion here as the authors state in the reply. This is incorrect. In fact, the argument used by the authors that 'general non-auditory readers could understand the concept' is not good, because it could indeed generate more confusion. A non-auditory reader may get different terms meaning different notions that are mixed. A tuning curve is what it is... and a tuning curve is exactly as the authors have understood and I mentioned in the original review. 'our understanding is that 'tonal receptive fields' are constructed from various tone frequencies and intensities'

This is a critical point and I cannot see the problem to say soothing that is correct. Furthermore, as I also mentioned and authors have considered, it maybe that the effects they describe do not occur a lower intensity levels, so I think that for the sake of transparency and accuracy, tuning curves should be removed unless authors really record from different combinations of intensities and frequencies so they can indeed have constructed from various tone frequencies and intensities. There are several examples in the literature where level affect different functions.

Reviewer #2 (Remarks to the Author):

I have no further concern.

Response to Reviewers

Reviewer 1

I would like to thank the authors for the effort and the good job , and I am generally happy with the revision, and most of my comments (although not all) have been taken care and addressed.

But the issue of tuning curves is simply incorrect. Authors try to convince the reader that tuning curves is equal to iso-intensity functions. I am sure they do this with their best intention, but in my view this is more harmful than helpful for the general audience

I don't think there is a confusion here as the authors state in the reply. This is incorrect. In fact, the argument used by the authors that 'general non-auditory readers could understand the concept' is not good, because it could indeed generate more confusion. A non-auditory reader may get different terms meaning different notions that are mixed. A tuning curve is what it is... and a tuning curve is exactly as the authors have understood and I mentioned in the original review. 'our understanding is that 'tonal receptive fields' are constructed from various tone frequencies and intensities'

This is a critical point and I cannot see the problem to say soothing that is correct. Furthermore, as I also mentioned and authors have considered, it maybe that the effects they describe do not occur a lower intensity levels, so I think that for the sake of transparency and accuracy, tuning curves should be removed unless authors really record from different combinations of intensities and frequencies so they can indeed have constructed from various tone frequencies and intensities. There are several examples in the literature where level affect different functions.

We would like to thank this reviewer for further clarification of the incorrect wording in our original manuscript. Following this suggestion, we have now replaced all the 'tuning curves' with 'iso-intensity functions.' There were three locations in the original manuscript, one each in the main text, main figure legend, and supplementary figure legend. The changes in the main text are highlighted in red in the revised manuscript. We apologize for the confusion that our wording may have caused, and we hope these edits resolve the issue.

Reviewer 2

I have no further concern.

We are glad to hear that all the concerns raised by Reviewer 2 have been addressed. We would like to thank both reviewers again for their constructive comments that made our manuscript more accurate and rigorous.

REVIEWERS' COMMENTS

Reviewer #1 (Remarks to the Author):

Authors have addressed all my comments, and I recommend acceptance as it is thanks a lot for taken care of them

congrats on your nice paper

best wishes